# *In vivo* chromatic and spatial tuning of foveolar retinal ganglion cells in *Macaca fascicularis*

**Tyler Godat**[1,2]*, **Nicolas P. Cottaris**[3], **Sara Patterson**[1], **Kendall Kohout**[1,2], **Keith Parkins**[1], **Qiang Yang**[1], **Jennifer M. Strazzeri**[1,4], **Juliette E. McGregor**[1], **David H. Brainard**[3], **William H. Merigan**[1,4], **David R. Williams**[1,2,4]*

**1** Center for Visual Science, University of Rochester, Rochester, NY, United States of America, **2** Institute of Optics, University of Rochester, Rochester, NY, United States of America, **3** Department of Psychology, University of Pennsylvania, Philadelphia, PA, United States of America, **4** David & Ilene Flaum Eye Institute, University of Rochester Medical Center, Rochester, NY, United States of America

\* tgodat@ur.rochester.edu (TG); david.williams@rochester.edu (DRW)

**Data Availability Statement:** The minimal data set for this manuscript is freely available with no restrictions, including the full set of underlying raw cellular responses to all stimuli, processed data

## Abstract

The primate fovea is specialized for high acuity chromatic vision, with the highest density of cone photoreceptors and a disproportionately large representation in visual cortex. The unique visual properties conferred by the fovea are conveyed to the brain by retinal ganglion cells, the somas of which lie at the margin of the foveal pit. Microelectrode recordings of these centermost retinal ganglion cells have been challenging due to the fragility of the fovea in the excised retina. Here we overcome this challenge by combining high resolution fluorescence adaptive optics ophthalmoscopy with calcium imaging to optically record functional responses of foveal retinal ganglion cells in the living eye. We use this approach to study the chromatic responses and spatial transfer functions of retinal ganglion cells using spatially uniform fields modulated in different directions in color space and monochromatic drifting gratings. We recorded from over 350 cells across three *Macaca fascicularis* primates over a time period of weeks to months. We find that the majority of the L vs. M cone opponent cells serving the most central foveolar cones have spatial transfer functions that peak at high spatial frequencies (20–40 c/deg), reflecting strong surround inhibition that sacrifices sensitivity at low spatial frequencies but preserves the transmission of fine detail in the retinal image. In addition, we fit to the drifting grating data a detailed model of how ganglion cell responses draw on the cone mosaic to derive receptive field properties of L vs. M cone opponent cells at the very center of the foveola. The fits are consistent with the hypothesis that foveal midget ganglion cells are specialized to preserve information at the resolution of the cone mosaic. By characterizing the functional properties of retinal ganglion cells *in vivo* through adaptive optics, we characterize the response characteristics of these cells *in situ*.

## Introduction

In primates, the fovea is a specialized region of the retina characterized by a high density of cones at its center and a corresponding radial displacement of retinal cells and retinal

points, some raw data examples, pre-processed and intermediary data examples, and the full analysis pipeline (including all programs used) at https://osf.io/s9qw4/ (DOI 10.17605/OSF.IO/S9QW4). The full set of recorded .avi videos cannot fully be deposited in an online repository because it is prohibitively large (> 1 TB). Additional data can be made available to individuals or labs by contacting the corresponding author. Additional data can be temporarily shared piecemeal in online repositories or mailed on a hard drive depending on the size of the request. ISETBio which has core code for the simulations in the manuscript is publically available at: https://github.com/isetbio/isetbio ISETMacaque, which contains code that is specific for modeling these AO experiments is freely available at: https://github.com/isetbio/ISETMacaque.

**Funding:** This work was supported by The National Eye Institute https://nei.nih.gov/about through grants NIH EY007125 to TG, NIH F32EY0322318 to SP, NIH U01EY025497 to DRW, NIH EY031467 to DRW, NIH EY021166 to WHM, NIH 2P30EY001319-44 to DRW and WHM, and through an unrestricted grant to the Flaum Eye Institute from Research to Prevent Blindness https://www.rpbusa.org/rpb/grants-and-research/grants/institutional-grants/. This work was also supported by funding from Facebook Reality Labs to DHB. This material is also based on work supported by the Air Force Office of Scientific Research under award numbers FA-9550-22-1-0167 and FA-9550-22-1-0044 to DRW. The funders had no role in study design, data collection and analysis, decision to publish, or preparation of the manuscript.

**Competing interests:** I have read the journal's policy and the authors of this manuscript have the following competing interests: D.R.W. has patents with the University of Rochester for adaptive optics imaging of the retina: US patent #6,199,986 "Rapid, automatic measurement of the eye's wave aberration". US patent #6,264,328 "Wavefront sensor with off-axis illumination" and US patent 6,338,559 "Apparatus and method for improving vision and retinal imaging". Q.Y. has patents with the University of Rochester, Canon Inc. and the University of Montana, for image stabilization algorithms: US patent #9,226,656: "Real-time optical and digital image stabilization for adaptive optics scanning ophthalmoscopy", US patent # 9,406,133: "System and method for real-time image registration", US patent #: 9,485,383, "Imaging based correction of distortion from a scanner" and US patent #:9,454,084, "Light source modulation for a scanning microscope". D.H.B. is

vasculature. These specializations, along with a correspondingly expanded representation of the central visual field in cortex [1–3], contribute to the superiority of foveal vision relative to peripheral vision in spatial resolution and color discrimination [1, 4–21]. To date, in excess of 20 classes of retinal ganglion cells (RGCs) have been identified in macaque retina [1, 4, 6, 7, 13, 17–20, 22–52], reflecting a complex array of retinal computations that precede additional processing in the brain. Despite these advances, a complete understanding of these independent circuits and their role in vision and behavior remains elusive. Even the precise role and scope of function of the most numerous and commonly studied RGCs, the midget ganglion cells, remains controversial, and there has been relatively little study at the very center of the foveola.

Electrophysiological responses from primate RGCs have been studied *in vivo* with a microelectrode that penetrates the sclera [53, 54] and *in vitro* using a single microelectrode [6, 27] or microelectrode arrays [55, 56]. However, the majority of the recordings in primate have been made in peripheral retina, often 30 to 70 degrees from the fovea due to the fragility of the fovea. The survival of an excised fovea is time limited, and the delicate structure of the foveal pit is often strained when flattened, sometimes tearing crucial circuitry. While the anatomical basis for foveal circuitry has been studied extensively [7, 32, 34, 38, 57, 58] and there has been some *ex vivo* physiology of foveal RGCs, most individual cell classes present in the primate fovea have not been directly studied, especially at the most central locations. Retinal processing can also be inferred by recording from retinorecipient neurons in the LGN, but the foveal representation of the LGN is located within a relatively thin cell layer making such recordings challenging [59]. Moreover, it has been difficult to identify the exact foveal center *in vivo* and determining the precise retinal eccentricity of single cell recordings in LGN is difficult [5, 9, 60–62]. Furthermore, RGCs project to many different targets other than LGN, so studying these cells at the retinal level gives the best chance at providing a more complete sampling.

In this study, we use functional adaptive optics cellular imaging in the living eye (FACILE) [12, 16, 63, 64], a technique that utilizes an adaptive optics scanning light ophthalmoscope (AOSLO) to image fluorescence elicited from the activity of RGCs that express genetically encoded calcium indicators, such as those described by Chen et al. [65]. Simultaneously, the cone mosaic is imaged with infrared wavelengths to map the cone density profile at the fovea. The cone mosaic images have a relatively high signal to noise ratio (SNR) and are also used to track residual eye motion in the fluorescence channel which has a relatively low SNR. Knowing the cone density profile for an individual subject allows for a more precise modeling of receptive field structure and more precise motion correction. Using this technique, we can both image and stimulate the centermost foveolar cones with spatial or chromatic stimuli and optically record the activity of the ganglion cells that are served by those cones, characterizing the physiology of the innermost foveal primate RGCs *in vivo*. Indeed, we can return repeatedly to the same ganglion cells for weeks to months, allowing repeated functional measures of the same cells. With this approach, we can use data from an individual animal that includes the optics, the cone density profile, and the RGC responses to create a detailed model of the first stages of visual processing.

## Materials and methods

### Animal care

The macaques (*Macaca fascicularis)* were housed in pairs in an AAALAC accredited facility. They had free access to food and water, providing a complete and nutritious diet. To supplement their ordinary lab chow, animals were given various treats daily such as nuts, raisins, and a large variety of fresh fruits and vegetables. An animal behaviorist provided novel enrichment

an inventor on US patent application 16/389,942. The authors also declare the following commercial funding sources that constitute additional competing interests: Q.Y. has undertaken consultancy work for Oculus VR and Boston Micromachine Corporation. W.H.M. has received research grants from the National Eye Institute and Research to Prevent Blindness but has no additional relevant declarations for these sources. D.H.B. has received research funding from the National Institutes of Health, Johnson & Johnson, and Facebook Reality Labs, but has no additional relevant declarations for these sources. D.R.W. has received research funding from the National Eye Institute, the Arnold and Mabel Beckman Foundation, Alcon but has no additional relevant declarations for these sources. D.R.W. has also received research funding from and undertaken consultancy work for Warby Parker. This material is also based on work supported by the Air Force Office of Scientific Research under award numbers FA-9550-22-1-0167 and FA-9550-22-1-0044. This does not alter our adherence to PLOS ONE policies on sharing data and materials. There are no restrictions on sharing of data and/or materials from this study.

items once per week, including items such as grass and treat filled bags, grapevines, and forage boxes. Daily enrichment included several pieces of manipulata: mirrors, puzzle feeders rotated among the animals, daily movies and/or music, as well as rotating access to a large free-ranging space with swings and elevated perches during any long periods between imaging sessions. All macaques were cared for by the Department of Comparative Medicine veterinary staff, including four full-time veterinarians, five veterinary technicians, and an animal care staff who monitored animal health and checked for signs of discomfort at least twice daily. This study was carried out in strict accordance with the Association for Research in Vision and Ophthalmoscopy (ARVO) Statement for the Use of Animals and the recommendations in the Guide for the Care and Use of Laboratory Animals of the National Institutes of Health. The protocol was approved by the University Committee on Animal Resources of the University of Rochester (PHS assurance number: D16-00188(A3292-01)).

## Immune suppression

Two macaques (M2, M3) received subcutaneous Cyclosporine A prior to intravitreal injection and imaging. Blood trough levels were collected weekly to titrate the dose into a therapeutic range of 150–200 ng ml$^{-1}$ and then maintained at that level. M2 began immune suppression in March 2018 with 6 mg kg$^{-1}$, then stepped down a month later to 4 mg kg$^{-1}$ which was maintained until October 2019 when suppression was stopped following the completion of all experiments used in this manuscript. M3 started immune suppression in May 2019 at 6 mg kg$^{-1}$, then stepped down to 3.4 mg kg$^{-1}$ which continues at time of submission. M1 did not receive any immune suppression. Current best practice for primate imaging in our lab has all new animals receiving immune suppression for several weeks prior to injection, as we believe that corresponds to the best viral transfection rates.

## AAV mediated gene delivery to retina

Intravitreal injections were carried out in each animal as previously described [66]. In M1, *7m8-SNCG-GCaMP6f* [67] was injected into the right eye; *AAV2-CAG-GCaMP6s* was injected into the left eye of M2 and into the right eye of M3. All vectors were synthesized by the University of Pennsylvania Vector Core. Before the injections, the eyes were sterilized with 50% diluted Betadine, and the vector was injected into the middle of the vitreous at a location approximately 4 mm behind the limbus using a tuberculin syringe and 30-gauge needle. Following injection, each eye was imaged with a conventional scanning light ophthalmoscope (Heidelberg Spectralis) using the 488 nm autofluorescence modality to determine onset of GCaMP expression and to periodically monitor image quality and eye health. Animal M1 was injected in March 2017, and data from that animal in this study was taken in March 2019, two years post-injection. Animal M2 was injected in March 2018, and data from that animal in this study was taken in March 2019 (1 year post-injection) and July 2019 (1.3 years post-injection). Animal M3 was injected in June 2019, and data from that animal in this study was taken in September 2020 (1.25 years post-injection) and in January 2021 (1.5 years post-injection). Efficiency of vector transmission was not explicitly examined in this study and though animal M1 had the lowest mean fluorescence of the three animals, we cannot meaningfully comment on what percentage of that effect is due to either the lack of immune suppression, post-injection time period, or differences due to the vector used in that animal.

## Anesthesia and animal preparation

Anesthesia and animal preparation followed established lab protocols as previously reported [e.g. 16, 68] and were performed by a veterinary technician licensed by the State of New York

(USA). All monkeys were fasted overnight prior to anesthesia induction the morning of an imaging session. The animal was placed prone onto a custom stereotaxic cart where it remained for the duration of the imaging session. A Bair Hugger warming system was placed over the animal to maintain body temperature. Monitoring devices including rectal temperature probe, blood pressure cuff, electrocardiogram leads, capnograph, and a pulse oximeter, were used to track vital signs. Temperature, heart rate and rhythm, respirations and end tidal $CO_2$, blood pressure, $SPO_2$, and reflexes were monitored consistently and recorded every fifteen minutes. Pupil dilation was accomplished using a combination of Tropicamide 1% and Phenylephrine 2.5%. A full description of all medications used in anesthesia induction, pupil dilation, intubation, and animal recovery can be found in McGregor *et al.* [68].

## Adaptive optics imaging

Data were collected using an AOSLO system previously described in Gray et al. [63]. An updated diagram of the system is shown in S1 Fig. An 847 nm diode laser source (QPhotonics) was used as a beacon for a Shack-Hartmann Wavefront Sensor (SHWS) to measure optical aberrations in each animal's eye in real time, and a Deformable Mirror (ALPAO) was used to correct those aberrations in a closed loop. A 796 nm superluminescent diode source (Superlum) was focused on the photoreceptor layer of the retina to collect reflectance images at an approximately 2 Airy disk confocal pinhole (20 μm), with the images used for both placement of stimuli and later registration (motion correction) of fluorescence images. A 561 nm laser (Toptica) and three LEDs (Thorlabs, center wavelengths 420 nm, 530 nm, 660 nm) were used for visual stimulation of foveal cones. A 488 nm laser source (Qioptiq) was focused on the ganglion cell layer to excite GCaMP fluorescence, which was detected through a 520/35 nm emission filter at an approximately 2 Airy disk confocal pinhole (20 μm) for animals M2 and M3 and an approximately 7.5 Airy disk confocal pinhole (75 μm) for animal M1, each of which was chosen to balance the tradeoff between signal strength and axial resolution. The poorer expression of GCaMP and consequently worse SNR in M1 motivated using the larger confocal pinhole for imaging in that animal. The 488 nm excitation light was presented only during forward scans and filled only the portion of the imaging window where ganglion cells were present to avoid exposing foveal photoreceptors and biasing stimulation. The 488 nm excitation light intensities on the retina were 3.4 mW cm$^{-2}$ in M1, 3.2 mW cm$^{-2}$ in M2, and 1.7 mW cm$^{-2}$ in M3, as presented through a dilated pupil size of approximately 6.7 mm for each animal. Retinal subtenses were 780 x 570 μm in M1, 765 x 560 μm in M2, and 740 x 540 μm or 505 x 380 μm in M3. Imaging sessions using the AOSLO took place starting around 9 am and lasted between two to four hours. Room lights were turned off during experiments so that ambient light was minimized and the animal was only exposed to the sources used in the AOSLO. Previous light imaging history was limited to clinical fundus and SLO imaging as mentioned in the AAV mediated gene delivery section.

**Light safety.** The total retinal exposure to all light sources was calculated for each proposed retinal location before each imaging session after measuring the optical power of each source at the pupil plane (sources measured as total optical power across the entire ~6.7 mm pupil: nominally 488 nm at 7–15 μW, 561 nm at 5 μW, 796 nm at 250 μW, 847 nm at 30 μW, and combined 5 μW from the LEDs at 420 nm, 530 nm, and 660 nm). For all animals, the total exposure was kept below the maximum permissible exposure for human retina according to the 2014 American National Standards Institute [69–71]. Most exposures were also kept below a reduced ANSI limit that was further scaled by the squared ratio of the numerical aperture of the human eye to the primate eye (~0.78) [72], though imaging sessions in M3 at the smaller FOV exceeded this reduced limit. Imaging sessions were at least five days apart, so

accumulated exposure was not taken account into the calculations. According to the ANSI standard, the maximum permissible exposure limit is expressed as a sum of ratios of source powers such that the limit value is unity. In this study, exposures ranged from values of 0.6–0.98 corresponding to 60%-98% of the exposure limit.

**Methods to maximize optical quality and signal across experiments.** The best axial resolution that can be achieved with the confocal AOSLO required for our experiments is 25–30 microns, approximately the diameter of 2–3 RGC somas [27], which means that in areas where the RGC layer is thick, such as the fovea [26, 46], there may be some optical crosstalk from multiple cells contributing to the recorded fluorescence signal. Therefore, as part of the validation of our method and to be certain in describing the physiology of individual cells, we operate in two distinct modalities. The first modality is a wide imaging field of view (FOV) of 3.70 x 2.70 deg where we can image hundreds of RGCs across a breadth of soma locations (in M1, M2 and M3), but not be certain that each cell is free of crosstalk. The second modality (only used in M3) is a smaller imaging FOV of 2.54 x 1.92 deg that is centered and focused on the innermost foveal slope, where the RGCs are in a monolayer—in this modality we can only image tens of cells at a time, but we are certain that the signal is not contaminated by optical crosstalk. We present wide field data from all three animals to show cell populations and general physiology, while we present data from the smaller field only from animal M3 to draw conclusions about the physiology of individual cells—the very innermost foveolar RGCs driven by the centermost cones. To ensure the desired imaging FOVs were used in each experiment, the AOSLO system was calibrated using Ronchi gratings of known spacing (80 lp/mm) to translate scanner control voltages into FOV measurements in degrees of visual angle.

To correct for longitudinal chromatic aberration between the cone reflectance source (796 nm), visual stimulus laser (561 nm), and GCaMP excitation laser (488 nm), the axial source and detector positions of the visible channels were optimized by initial calculation using chromatic dispersion data in humans [73] scaled for the macaque eye. Lateral detector positions were optimized first in a model eye and then further optimized empirically *in vivo* using algorithms such as simplex [74] to iterate over lateral pinhole positions and find the location with maximum detector mean pixel value. For the data in M3 at the inner edge of the foveal slope, an additional experiment optimized the responses of 37 RGCs to 28 c/deg drifting gratings at 6 Hz to find the best axial source position of the 488 nm excitation laser, to account for RGCs on the foveal slope differing in axial position. Transverse chromatic aberration was not critical to stimulus placement in these experiments, but lateral positions of confocal detection pinholes were also optimized using similar algorithms [75].

Perhaps as a side effect of the *in vivo* viewing eye being held open for long periods of time during imaging sessions, optical quality tended to degrade over the course of the experiment, slowly at first, but more quickly after the first hour of imaging. To mitigate this, all grating stimuli were presented at the beginning of a session when optical quality was the highest, and only for the first 30 minutes, to minimize degradation of high spatial frequency responses.

Lastly, the isoplanatic patch size of our system is approximately 1–1.5 deg, which is smaller than both of the imaging fields (3.70 x 2.70 deg and 2.54 x 1.92 deg) used in this study. This degrades quality across the entire image and produces field aberrations towards the edges of the images where some ganglion cells were imaged or cones were stimulated. This anisoplanatism, coupled with the fact that the wavefront correction is at a different wavelength than stimulation, means that there could be defocus (see Modeling section) or higher order residual aberrations that preferentially blur high spatial frequency gratings. Care was taken to mitigate or optimize these sources of residual blur, but they cannot completely be eliminated with our current instrument. Therefore, the spatial frequency responses we measured represent a lower bound for the cell responses *in situ* when all sources of residual blur are removed.

**Chromatic stimuli.** To drive the chromatic responses of RGCs connected to foveal photo-receptors, we presented a 1.3 deg diameter spatially uniform, circular LED stimulus (Thorlabs, center wavelengths and full-widths-at-half-maximums 420±7 nm, 530±17 nm, 660±10 nm) through a Maxwellian view system [76] (S1 Fig). The stimulus was sinusoidally modulated in time. Mean intensities on the retina and temporal frequencies were 6.8 mW cm$^{-2}$ at 0.2 Hz in M1, 7.0 mW cm$^{-2}$ at 0.2 Hz in M2, and 7.6 mW cm$^{-2}$ at 0.15 Hz in M3. The slow temporal frequencies were chosen to accommodate the slow temporal response of the GCaMP6 calcium indicator [65] and were adjusted by 0.05 Hz in M3 to avoid potential confound with the respiration rate in that animal. The LED primary wavelengths were chosen to maximize excitation of the L, M, and S cone photopigments [77] while minimizing significant confound by macular pigment absorption [78, 79], the spatial distribution of which was not measured in this study. Each stimulus presentation was 90 s long, following a 30 s adaptation period in which the mean luminance white point and excitation laser were on. The cone fundamentals were constructed using the Govardovskii standard template [80, 81] and the wavelengths of peak absorption for macaque primates [82]. The transmissivity of the eye (including cornea, aqueous, lens, and vitreous) was modeled using measured data from the rhesus macaque [83]. To generate the luminous efficiency function, an equal weighting of the L and M cone fundamentals was used, as the ratio of L and M cones in macaques is close to unity on average [84]. The LEDs were calibrated using a spectrometer (Ocean Optics) validated by a NIST traceable blackbody source. Five silent substitution [85] stimuli were presented at the foveal center while GCaMP responses were recorded. The Psychophysics Toolbox [86] for MATLAB was used to calculate the appropriate power modulations needed: a nominally isoluminant stimulus targeting both L and M cones in counterphase (L modulation 15%, M modulation 17%), an L cone isolating stimulus (24% modulation), an M cone isolating stimulus (33% modulation), an S cone isolating stimulus (92% modulation), and an achromatic luminance stimulus (100% modulation, all cone classes). There was also a control stimulus where the adaptation period was followed by continued presentation of the mean luminance white point. Within one imaging session, each of the stimuli used in that session was presented three times and responses were averaged within the experiment. In M1, only one experiment was performed. In M2 and M3, two experiments across the entire width of soma locations (large imaging FOV) were performed and responses across sessions were averaged. In M3, three separate experiments at the innermost foveal RGCs (small imaging FOV) were performed and averaged.

**Grating stimuli.** To probe the spatial configuration of foveal RGC receptive fields, we presented monochromatic (561 nm) horizontally-oriented sinusoidal drifting gratings at 6 Hz and 100% contrast. The grating stimuli were presented within the scanned imaging FOV and occupied a square subregion that was either 1.9 deg (large imaging FOV) or 1.3 deg (small imaging FOV) in extent. Mean light intensities on the retina were 1.72 mW cm$^{-2}$ in M2 (large imaging FOV), 1.8 mW cm$^{-2}$ in M3 (large imaging FOV), and 3.2 mW cm$^{-2}$ in M3 (small imaging FOV) imaging only the innermost RGCs closest to the center of the foveola. No data from drifting gratings is shown for animal M1, as the signal was too weak and noisy to produce reliable data from these stimuli. The temporal frequency of these gratings was too fast to allow GCaMP6 to track modulation, so cellular responses were characterized by quantifying the increase in steady fluorescence during stimulation, as opposed to using fluorescence modulation as was done for the slower chromatic stimuli. Grating stimuli were 45 s in duration and were preceded by 15 s adaptation to both the mean luminance of the 561 nm stimulus laser and the 488 nm imaging laser.

In the small imaging FOV used with M3, the increased resolution meant that the spatial frequency range was 4 c/deg to 49 c/deg. A control (spatially uniform mean luminance) and 14 gratings of different spatial frequency were presented twice each within a session and

responses were averaged. Gratings were presented in decreasing order of spatial frequency, to minimize reduction in high spatial frequency response related to decreasing optical quality over the course of the experiment. There were three sessions at the small FOV in M3, and the variability and averages are shown in the results. We note that as gratings were only presented in one orientation, the responses of some cells might be modified by orientation selective effects [87].

In M2 and M3 at the large imaging FOV, a control and 14 gratings varying in spatial frequency from 2 c/deg to 34 c/deg were presented. Each grating was presented twice within a session and responses were averaged for one session in M2 and across two sessions in M3. These data are not presented in the results because they did not include high enough spatial frequency responses that permit the modeling used in this work (S5 Fig).

## Data analysis

To remove blurring and other effects of residual eye movements present in the raw recordings, each video frame of the fluorescence recording was co-registered using corresponding frames from a high signal-to-noise reflectance video of the photoreceptor layer at the same retinal location. During imaging, a reference frame was taken in the reflectance channel to allow for real-time stabilization; this reference frame was also used as the reference for the frame-to-frame cross-correlation method image registration [88] of all videos taken at that location. One registered video, typically one of the videos corresponding to the peak spatial frequency response (usually 10–30 cycles/deg), was temporally summed to create a high SNR fluorescence image of the ganglion cell layer, and individual ganglion cells were segmented using the open-source software GIMP. Typically, all identifiable cells in the focal plane were segmented, though cells were excluded if the boundaries between two or more adjacent cells were unclear. The segmentation mask from GIMP was exported to MATLAB, where it was applied to all registered videos at that location to isolate data from individual RGCs. For every frame, the mean of the signal within each cell mask was computed to produce an average signal time course for each cell. When the same stimuli were repeated within a session, the signal time courses for each cell were averaged across all repeated trials.

For the chromatic data, the temporal speed of the flicker (0.15–0.20 Hz) was well within the temporal resolution of the GCaMP6, thus cells responding to the stimulus exhibited a quasi-sinusoidal modulation in fluorescence level where the amplitude and phase depended on the receptive field properties of the cell and the frequency of the response depended on the stimulus frequency. Each cell's time course was filtered using a Hann windowing function before being Fourier transformed into the frequency domain. The Fourier amplitudes were normalized by subtracting the mean noise at higher frequencies (0.32 Hz—1.08 Hz) and then dividing by the standard deviation of the noise to produce a response metric equivalent to the sensitivity index d'. The metric was computed for each stimulus for every cell, so that the cells could be plotted in two different configurations from the literature [e.g. 60]. In animals M1 and M2, where an L-M nominally isoluminant stimulus, an S-isolating stimulus, and an achromatic/luminance stimulus were used, no further analysis was needed, and the data was simply plotted along S vs. L-M axes. In animal M3, since single cone isolating stimuli were used for L, M, and S cones, it was possible to reproduce cone weighting plots such as those in Derrington *et al.* [60]. As in that paper, we calculated the cone weights by dividing the response metric by the cone modulation from the stimuli and normalized them all such that the sum of all cone weights added to unity.

For the spatial frequency data, the temporal speed of the drifting gratings (6 Hz) exceeded the temporal resolution of the GCaMP6s indicator [65], thus cells responding to the grating

exhibited an overall increase in fluorescence that plateaued after approximately 10–20 s. The background level for each cell was calculated as the mean of the first 10 s of fluorescence during the adaptation period, while the signal was calculated as the mean of the last 10 s of fluorescence recorded during the presentation of the stimulus. The ΔF/F metric commonly used in calcium imaging was calculated for each spatial frequency as the background response at that frequency subtracted from the signal at that frequency, all divided by the signal at 17 c/deg, a spatial frequency producing a robust response across the majority of the cells. This function of ΔF/F with respect to spatial frequency is what we refer to as a spatial transfer function (STF) for each RGC measured. An error metric was also calculated for each ΔF/F measurement by using the partial differential error propagation formula using variances [89] and the measured variances in the background response, the signal response, and the signal at 17 c/deg.

## Modeling of photoreceptor mosaic and RGC response characteristics

To connect the measured spatial transfer functions (STFs) to the receptive field (RF) organization of the underlying RGCs, we employed a computational model that simulates optical, spectral, spatial, and temporal components of the AOSLO stimulation apparatus, as well as the animal's optics and cone mosaic structure. The model computed cone mosaic responses to simulations of the stimuli used to measure the STF and derived an STF model fit for the RGC under study. The model assumed that cone signals are pooled linearly and instantaneously (ignoring temporal pooling dynamics) by center and surround mechanisms of an RGC according to a difference of Gaussians (DoG) [90] spatial profile. The parameters of the DoG (and therefore the cone pooling weights) were estimated by minimizing the error between the model STF and the measured *in vivo* fluorescence-based STF. A model scenario where the center mechanism's Gaussian weighting was replaced by center weighting on just a single cone was also considered. The modeling pipeline depicted schematically in the results, was implemented within the Imaging Systems Engineering Tools for Biology (ISETBio) software framework [91, 92].

The drifting monochromatic sinusoidal gratings used to measure RGC STFs via the AOSLO apparatus were modeled as temporal sequences of ISETBio spatial-spectral radiance scenes, where each scene models one frame of the displayed stimulus. The spectral characteristics, spatial extent, and the temporal properties of the AOSLO display subsystem were taken into account in generating the ISETBio scenes. The spectral profile of the monochromatic beam was modeled as Gaussian shaped with a peak at 561 nm and a full-width-half-max (FWHM) of 5 nm with a mean irradiance on the retina of 1.29 mW cm$^{-2}$. The power estimate included absorption by the lens pigment. The visual stimulus as imaged on the retina had a pixel size of 1.03 μm (0.005 deg) with spatial extent 140 x 140 μm (0.7 x 0.7 deg). All sinusoidal gratings were modeled with a nominal contrast of 1.0, drifting at 6 Hz with a refresh rate of 25.3 Hz for a duration of 666 ms (4 cycles).

The animal's optics during AOSLO-based stimulation were modeled as a diffraction-limited optical system with a small amount of residual defocus blur. The diffraction limit was obtained using the 6.7 mm pupil diameter employed in the experiment, assuming a conversion between degrees of visual angle and retinal extent of 199 μm/deg (calculated using the ratio of the axial length of M3 16.56 mm to the model human 24.2 mm multiplied by the model human conversion of 291.2 μm/deg). Residual blur in the AOSLO, which might occur due to a slight defocus of the stimulus with respect to the plane of cone inner segments in the retina, was modeled by adjusting the Zernike defocus coefficient used to compute the optical point spread function. Residual blur could also be generated by any other aberrations in the system, imperfect AO correction, or other sources, so the Zernike defocus coefficient was used as a

proxy for all sources of residual blur. The amount of residual blur was not known *a priori* and was estimated as part of model fitting as described below. The temporal sequence of ISETBio spatial-spectral radiance scenes that model the AOSLO stimulus were passed via the simulated optical system to generate a corresponding sequence of spatial-spectral retinal irradiance images, which were processed by the cone mosaic model as described next.

An ISETBio model of the animal's cone mosaic was generated from cone density maps measured during AOSLO imaging, using an iterative algorithm described previously [91]. The spatial extent of the modeled cone mosaic was 1.3 x 1.3 deg, with a maximal cone density of 270,200 cones mm$^{-2}$ with relative L:M:S cone densities of 0.48:0.48:0.04 (quasi-regular S cone packing, random L and M cone packing), and without a tritanopic (S-cone free) area at the fovea. Cones were modeled with Gaussian entrance apertures with a characteristic radius equal to $0.204 * \sqrt{2} * D$, where $D$ is the inner segment diameter measured during AOSLO imaging [10]. In this cone mosaic model, cone outer segment lengths and macular pigment all varied with eccentricity [91], and the cone quantal efficiencies were based on the Stockman-Sharpe (2000) normalized absorbance measurements [93, 94]. The modeled macular pigment density is taken as the human values provided by Stockman *et al.* (1999) [93] (http://www.cvrl. org/database/text/maclens/macss.htm), and the modeled lens pigment density is for young human subjects ($< 20$ years) as reported by Pokorny *et al.* [95].

To compute a cone's excitation, the spatial-spectral irradiance impinging on the retina was first spectrally weighted by the product of macular pigment transmittance and by each cone's spectral quantal efficiency, subsequently integrated over wavelength, and spatially integrated over the cone's Gaussian aperture. This excitation response was integrated over the temporal duration of each stimulus frame (39.5 ms), to estimate a spatial map of the expected excitation events count $E^j(\omega,t)$, for the $j$-th cone in the mosaic, at time $t$, in response to a drifting grating of spatial frequency $\omega$. In these simulations, we did not include Poisson noise in $E^j(\omega,t)$, nor did we introduce positional jitter between the cone mosaic and the retinal stimulus due to fixational eye movements. In the measured AOSLO data, the animals were anesthetized with eye muscles paralyzed, and digital tracking was employed for stimuli, so retinal motion relative to stimuli was minimized.

Assuming that cones are adapted to the mean background irradiance, the spatiotemporal excitation response $E^j(\omega,t)$ was converted to a spatiotemporal modulation response, $R^j(\omega,t)$, by first subtracting the excitation of the cone to the background stimulus $E_0{}^j$, and then dividing by it separately for each cone $j$, i.e.:

$$R^j(\omega, t) = \frac{E^j(\omega, t) - E_0^j}{E_0^j}$$

This operation captures in broad strokes an important effect of the photocurrent generation process which converts cone absorption events in the inner segment into ionic currents flowing through the cone outer segment, and which in effect down-regulates the stimulus-induced cone excitation rate with respect to the background cone excitation rate.

Model ganglion cell responses $RGC(\omega,t)$, were computed from the cone contrast responses by weighting responses with corresponding center and surround cone weights, followed by spatial pooling within the antagonistic center and surround mechanisms as follows:

$$RGC(\omega, t) = RGC_c(\omega, t) - RGC_s(\omega, t)$$

$$= \sum_j W_c^j * R^j(\omega, t) - \sum_j W_s^j * R^j(\omega, t)$$

where $W_c^j$ and $W_s^j$ are the weights with which the center and surround mechanisms respectively pool the responses $R^j(\omega, t)$. We did not model temporal filtering or delay between center and surround responses. Although real RGC responses may be affected by both temporal filtering and a center-surround delay, measurements in this study only recorded the amplitude of responses at one temporal frequency with no response phase information, so we could not meaningfully estimate temporal RGC parameters.

To make computation of cone weights more tractable, we assumed that the spatial distribution of cone weights to the center and surround mechanisms had concentric, radially-symmetric Gaussian profiles (multi-cone RF center scenario), or that the center drew on a single cone with a Gaussian surround (single-cone RF center scenario):

$$W_c^j = \begin{cases} k_c * \exp\left[-\left(\dfrac{d_j}{r_c}\right)^2\right] & (multi - cone\ RF\ center\ scenario) \\ k_c & (single - cone\ RF\ center\ scenario) \end{cases}$$

$$W_s^j = k_s * \exp\left[-\left(\frac{d_j}{r_s}\right)^2\right]$$

where $d_j$ is the distance between cone-$j$ and the spatial position of the center mechanism of the model RGC (which is taken as the geometric centroid of the locations of the cones driving the center mechanism). The parameters $k_c$, $k_s$, $r_c$, and $r_s$, which represent the center and surround peak sensitivities and characteristic radii, respectively, of the DoG RF model were determined by minimizing the root mean squared error (RMSE) between the model-predicted STF, $STF^m(\omega)$, and the measured STF, $STF^{\Delta F/F}(\omega)$, accumulated over all spatial frequencies $\omega$:

$$RMSE = \sqrt{\frac{1}{N} * \sum_{\omega=\omega_1}^{\omega_n} \frac{\beta(\omega)}{\epsilon(\omega)} * \left[STF^m(\omega) - STF^{\Delta F/F}(\omega)\right]^2}$$

where $\epsilon(\omega)$ is the standard error of the mean of the $STF^{\Delta F/F}(\omega)$ measurement, and $\beta(\omega)$ is a high-spatial frequency boost factor which linearly increases with spatial frequency from 0.1 to 1.0 over the spatial frequency range 4.7 to 49 c/deg. This boost factor was introduced to emphasize high spatial frequency measurements, the stimulus regime which is maximally informative about the properties of the center mechanism.

In some cells, and for certain recording sessions, the measured $STF^{\Delta F/F}(\omega)$ dropped below zero for the lowest spatial frequencies. When this occurred, it was accompanied by an apparent overall downward shift in $STF^{\Delta F/F}(\omega)$ across all frequencies, so we assumed that in such cases the background fluorescence was overestimated. To compensate for this, we fit the model to $STF^{\Delta F/F}(\omega) - \min(STF^{\Delta F/F}(\omega))$ instead of $STF^{\Delta F/F}(\omega)$ in cases where $STF^{\Delta F/F}(\omega)$ dropped below zero for any spatial frequency.

To reduce the chance of the minimization algorithm getting stuck at a local minimum of the error function, we employed a multi-start minimizer which was run 512 times, keeping the results from the starting point with the minimum RMSE.

The model-predicted STF, $STF^m(\omega)$, was computed by fitting a sinusoidal function to $RGC^m(\omega, t)$,

$$A(\omega) * \sin\left[2\pi ft - \theta\right]$$

where *f* is set to the temporal frequency of the drifting gratings, and θ, A(ω) are free

parameters. At each spatial frequency, the amplitude of the fitted sinusoid, A(ω), was taken as STF$^m$(ω).

To interpret the measured STFs, we considered four different model scenarios: single cone RF centers with and without residual defocus in the AOSLO apparatus, and multiple cone RF centers with Gaussian weighting, with and without AOSLO residual defocus. For each scenario, the model was fit to the STF data of an examined RGC at multiple positions within the model cone mosaic and the final model was selected as the one with the minimum RMSE over the examined positions. The multi-position model fitting/selection was performed to take into account local inhomogeneities of the cone mosaic as exact RF center of each recorded RGC is only known to reside approximately within the central 40 microns. Details of the model fitting differences between each scenario, model training, and comparisons between the modeled cone mosaic and the measured mosaic from M3 are provided in S7–S10 Figs. All figures in the main body of the paper assume the single cone center with fixed residual 0.067 D defocus scenario, unless otherwise noted.

## Results

### *In vivo* functional imaging of foveal RGCs

In this study, optical stimulation of foveal cones elicits the activity of retinal ganglion cells in the living macaque eye. Intravitreal injection of an adeno-associated virus (7m8 or AAV2) with a promoter (ubiquitous CAG or RGC-specific) produced expression of the calcium indicator GCaMP6f [65] in animal M1 and GCaMP6s [65] in animals M2 and M3. As previously reported, injections in all three animals produced expression in a ring of RGCs surrounding and radiating out from the foveal avascular zone where the internal limiting membrane is thin [66, 68, 96] (Fig 1A). Stimuli were presented to the centermost foveal cones while a 488 nm imaging laser stimulated fluorescence from GCaMP expressing RGCs in a rectangular area placed either nasally or temporally relative to the cones. Fig 2A and 2B show the fluorescence response of a single RGC to a simple luminance flicker stimulus in time as well as the resulting Fourier transform, which shows the cell responding primarily at the stimulus frequency. Fig 2C shows that, under our stimulus paradigm, the imaging light did not stimulate foveal cones and confound the recorded functional responses after a 561 nm adapting light was presented to the foveal cones. In each animal, hundreds of RGCs were recorded simultaneously using this paradigm, and we used the precision of the AOSLO system to return to the same cells over multiple sessions spanning weeks to months. For examples of the raw time course responses of cells and the variability of responses to individual stimuli across experiments, see S2 and S3 Figs.

### Reliability and precision of foveal RGC recordings

Across multiple sessions, RGC somas were recorded from over a range of soma locations between 1.2 and 3.5 degrees from the center of the foveal avascular zone (FAZ). When averaging data across multiple experiments, only cells that were visible in all experiments were used. In M3, a follow-up experiment examining only cone density (Fig 1B) found the center of the FAZ to be roughly 41 microns inferior and 51 microns nasal from the peak cone density which agrees with the range of data reported from humans [14]. Based on detailed receptive field mapping performed by McGregor *et al.* [16], the cones driving the receptive fields of the cells examined in all three animals are likely to be located at eccentricities less than 36 arcmin from the foveal center, or a retinal radius of 120 microns calculated using the primate model eye [97] with the axial lengths of each animal (M1: 17.51 mm, M2: 17.20 mm, M3: 16.56 mm). According to the same mapping, the receptive fields of the innermost cells we recorded from

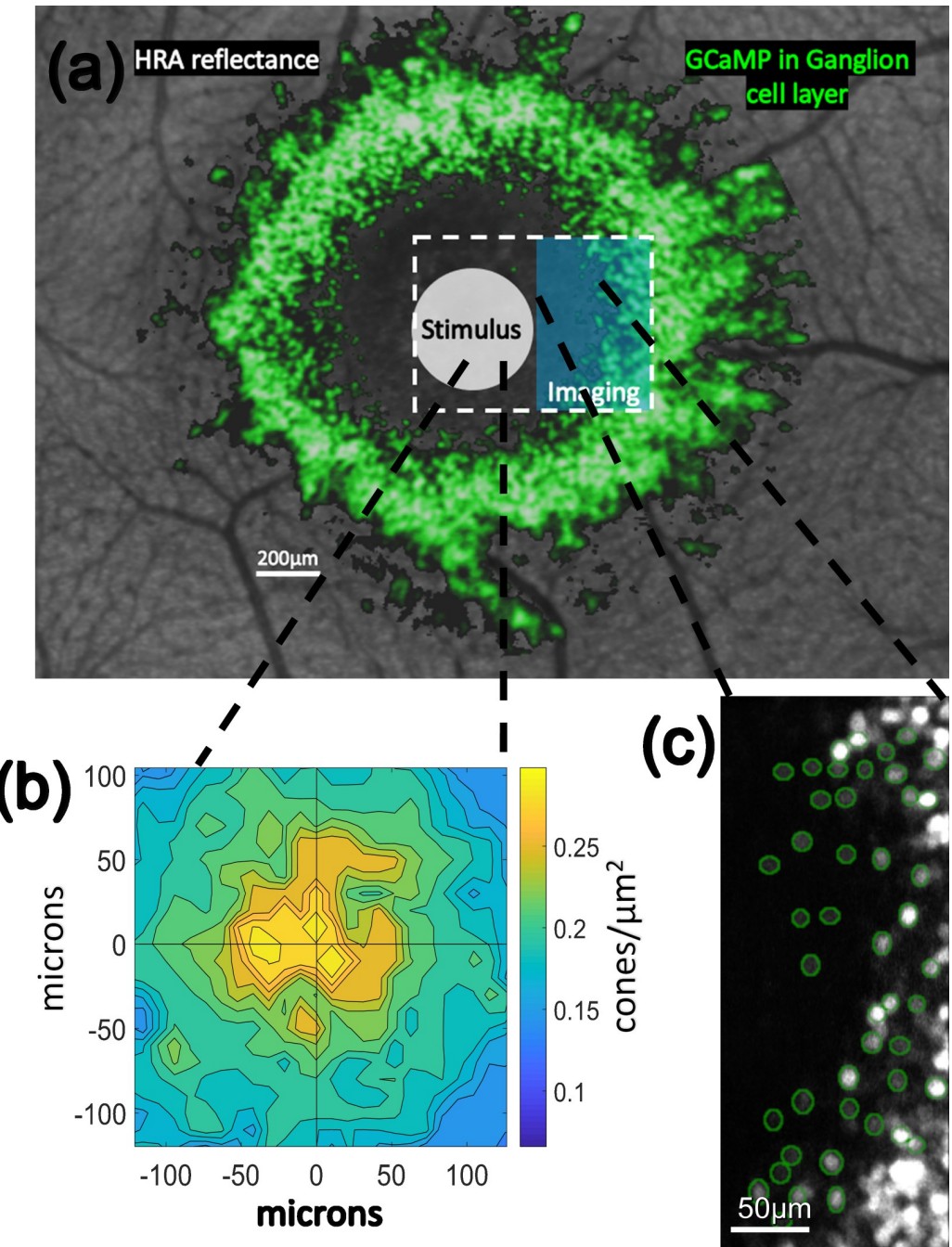

**Fig 1. Stimulus presentation and recording paradigm.** In **(a)** the general stimulation paradigm is shown. Background is an image of the retina of M3 using the blue reflectance imaging modality of a Heidelberg Spectralis instrument. In false color green are GCaMP-expressing cells from M3 imaged using the blue autofluorescence modality of the same Spectralis instrument. When using the AOSLO instrument, videos are captured within a rectangular field of view (white dashed line) while stimuli (white circle) such as cone-isolating flicker or drifting gratings (which were rectangular in shape and covered approximately the same area as the white circle) are presented to and centered on the centermost foveal cone photoreceptors. A 488 nm imaging laser in the AOSLO (blue rectangle) excites GCaMP-mediated fluorescent responses of ganglion cells surrounding the foveal slope. **(b)** shows the recorded cone densities at the fovea of M3 and **(c)** shows the ganglion cells segmented (green) at the innermost edge of the foveal slope in M3.

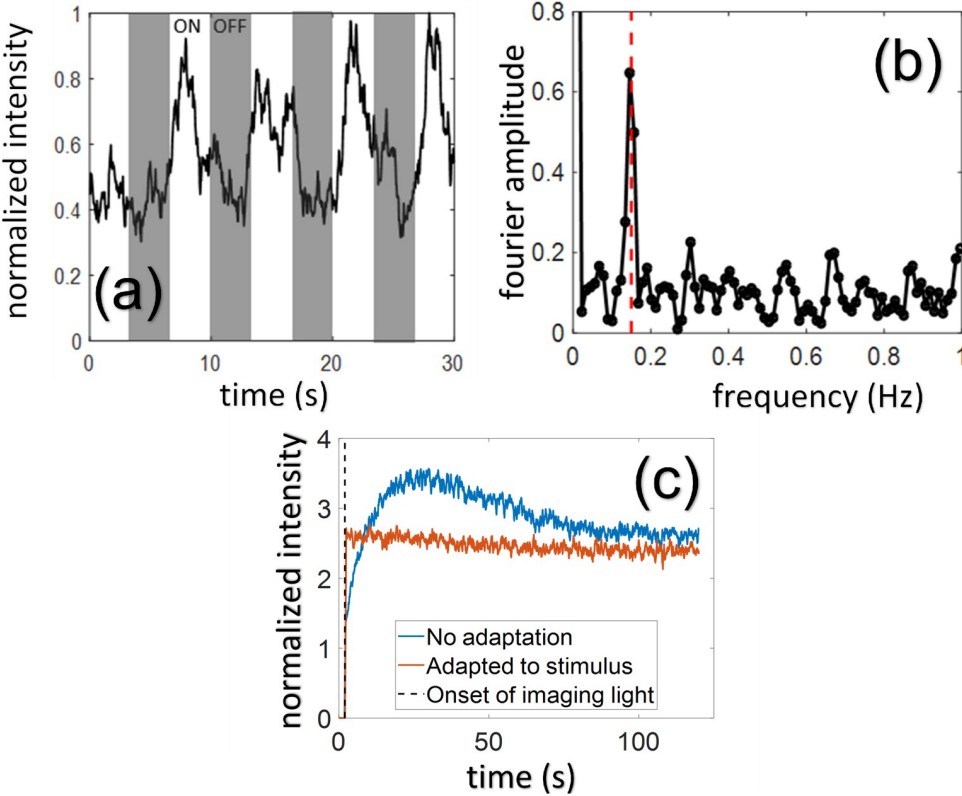

**Fig 2. Data collection from ganglion cells *in vivo*. (a)** shows the smoothed time course of a single RGC fluorescence response to a 0.15 Hz luminance flicker stimulus. **(b)** shows the amplitude of the Fourier transform of the response time course, recovering the response peak at the stimulus frequency of 0.15 Hz (red dashed line). **(c)** shows the effect of the imaging light on cone-mediated response to stimuli in 62 foveal RGCs. The 488 nm imaging laser which excites GCaMP fluorescence can scatter to foveal cones and confound cone stimulation. If the imaging light is turned on without adaptation to any light, there is a quick rise and slow falloff in fluorescence due to cone responses feeding RGCs. However, if the 561 nm stimulus laser (used for drifting gratings) or LED white point (used for chromatic stimuli) is presented at a mean power of 2–2.5 μW during a short 30s adaptation period prior to imaging onset, then there is no corresponding effect of the imaging light confounding the fluorescence signal. In all recordings, an adaptation light preceded each recording to prevent this transient effect of scattered light from the imaging laser.

in M3 (Fig 1C) are likely to be driven by cones found at eccentricities less than 6 arcmin, or a retinal radius of 20 microns calculated using the same primate model eye and the axial length of M3 (16.56 mm). All visual stimuli delivered to foveal cones were large enough to cover both the FAZ center and the location of maximum cone density and were some 10–60% larger than the cone area servicing even the furthest cells we recorded from, ensuring the best possible chance of recording responses from the maximum number of RGCs.

## Chromatic tuning of foveal RGCs

For functional classification of RGCs with receptive fields at the foveal center, chromatic tuning was measured across multiple imaging sessions. Amplitude and phase responses (Fig 3A and 3D) to spatially uniform 0.15 Hz or 0.20 Hz flicker modulations that were directed along a combination of L, M, L-M, S, and achromatic/luminance directions in color space enabled reliable identification of chromatic tuning of RGCs in all three animals. For each cell, the raw fluorescence time course was measured in response to each visual stimulus, and the Fourier transform was used to find the cell's response amplitude at the stimulus frequency. The signal-

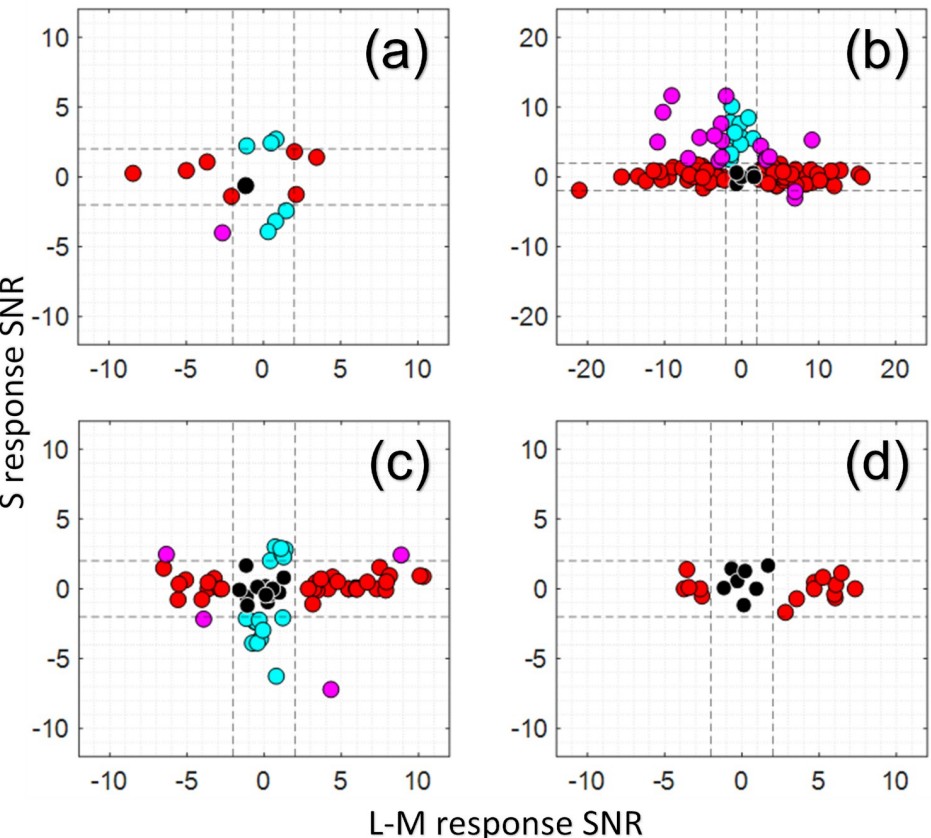

**Fig 3. Chromatic response properties of RGCs from three different animals.** Black dots show luminance only responding cells, red dots show L-M and M-L cone opponent cells, blue dots show S responding cells, while magenta dots show cells that responded to both L-M and S stimuli. Dotted lines show the significance cutoff of SNR = 2 calculated based on cell responses to the control stimulus (see Methods). Cell response is captured by our signal-to-noise metric described in the Methods. Response phases were also calculated but are not directly shown; however, positive SNRs correspond to responses that were in phase with the stimulus (ON response), while negative SNRs correspond to responses that were approximately 180 deg out of phase with the stimulus (OFF response). The luminance axis is perpendicular to the plane shown and is not included for readability and because not all cells responded to the luminance stimulus. Luminance only cells shown in black thus include both ON and OFF cells. In **(a)**, responses from 15 cells in M1 are shown after presentation of spatially uniform luminance, L-M isoluminant and S-isolating stimuli at 0.2 Hz. In **(b)**, responses from 126 cells in M2 are shown after presentation of spatially uniform luminance, L-M isoluminant and S-isolating stimuli at 0.2 Hz. In **(c)**, responses from 83 cells (a subset of the cells in Fig 4A, does not contain any cells from Fig 3D) in M3 are shown after presentation of spatially uniform luminance, L-isolating, M-isolating, and S-isolating stimuli at 0.15 Hz. In **(d)**, responses from 22 cells (a subset of the cells in Fig 4D) serving the very central fovea of M3 are shown after presentation of spatially uniform luminance, L-isolating, M-isolating, and S-isolating stimuli at 0.15 Hz. These cells are closer to the foveal center than any of the cells shown in (c). In **(c-d)**, all stimuli were single-cone isolating, so the L-M axis is calculated from the single cone L-isolating and M-isolating responses and their relative phase.

to-noise (SNR) response metric was calculated as the response Fourier amplitude (signal) minus the mean amplitudes of higher spatial frequencies (noise), all divided by the standard deviation of the noise. This analysis revealed several functional groups including L-M and M-L opponent cells, S-ON and S-OFF cells, ON and OFF luminance cells, and cells with mixed L vs. M ± S responses. In M3, single cone isolating stimuli were presented to the fovea and the individual cone weights feeding RGCs were calculated as was done previously in LGN [60] (Fig 4A and 4B). This analysis estimates the relative contribution of each cone class to the response of a cell and revealed further diversity of functional groups, including cells with

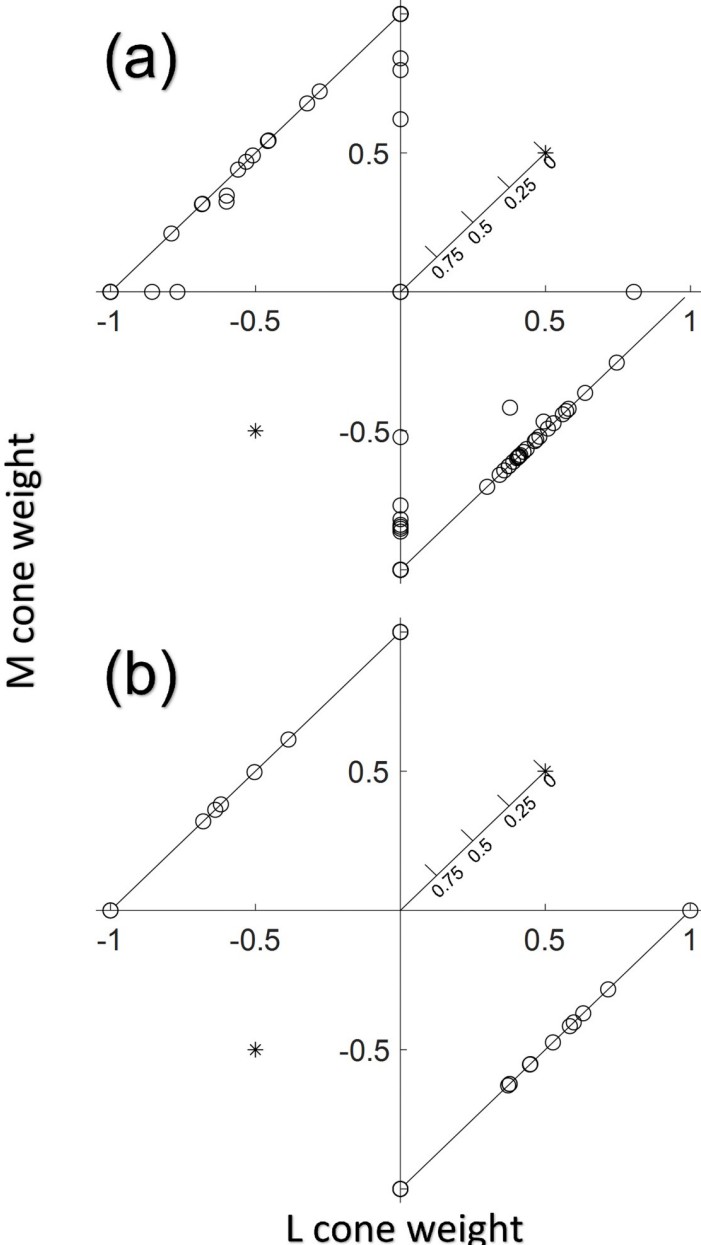

**Fig 4. Cone weights of RGCs from M3.** The horizontal axis shows L cone weight, the vertical axis the M cone weight, and the distance from the origin shows the S cone weight where further away from the origin corresponds to less S-cone weighting. In (**a**), 100 RGCs (a superset of the cells in Fig 3C, does not contain any cells from Fig 4B) were recorded across the entire width of the GCaMP-expressing ring in M3. There were 3 cells that only showed S cone responses, represented by overlapping circles at the origin. There were also 4 L+M and 11 -L-M luminance only cells that are represented by asterisks in the upper right and lower left. Because these cells only responded to the luminance stimulus but not the L-isolating or M-isolating stimulus, the response sign is known but not the exact L or M cone weighting. In (**b**), 34 cells (a superset of the cells in Fig 3D) at the innermost edge of the foveal slope in M3, served by the centermost foveal cones, were recorded from. No consistent S only responses were observed, and there were 5 L+M and 2 -L-M luminance only cells.

responses to only L or M cones, as well as various S cone connections to cells with either L or M cone input that were not L vs. M opponent. In figures with individual cell labels for putative midget ganglion cells, cells are named according to their suspected center cone (e.g. L1, L2,

M1, etc.). The suspected center cone was chosen based on the assumption that the center response would be the larger than the surround, and that midget ganglion cells connect to a single L or M cone in their center; this labeling is merely a convenience and should not be taken as a definitive measurement of the center cones.

### Identification of foveolar midget RGCs using chromatic tuning

Across all three animals, cells with L-M or M-L cone opponent responses (and no response to S-cone stimuli) were identified as putative midget RGCs. There were a large number of these cells as can be seen in Figs 3 and 4: the proportions of these putative midget cells were 44% (95% CI true proportion of 44±17%) of responsive RGCs closest to the fovea (1.2–1.8 deg soma locations) in M3 (small imaging FOV), 33% (95% CI true proportion of 33±9%) of responsive RGCs over the entire measured range in M3 (large imaging FOV), 72% (95% CI true proportion of 72±8%) of responsive RGCs in M2 (large imaging FOV), and 47% (95% CI true proportion of 47±25%) of responsive RGCs in M1. It is expected that midget RGCs comprise some 70–90% of foveal cells [20, 51, 98], and the confidence intervals in M1 and M2 contain values in that range. In M3, single cone isolating stimuli were used as detailed in the chromatic stimuli section—it is expected that these stimuli produce lower responses in the classical cone opponent model of midget RGCs by not driving the center and surround in counterphase as would the isoluminant L-M stimulus used in M1 and M2. There were many cells in M3 (an additional 35% of the closest foveal RGCs, and an additional 32% over the entire measured range) with only an L cone response or only an M cone response and no S cone response. It is possible that some of these cells are cone opponent, but that the single cone isolating stimuli produced too small of a response from the antagonistic surround in those cases. Based on this uncertainty, we cannot say for sure whether the observed number of putative midget RGCs in M3 is outside the range of expected values.

Soma size is a useful anatomical measure for distinguishing more peripheral ganglion cell types and was examined here as a potential correlate with our putative midget RGCs. RGC soma sizes were measured by two different observers (S4 Fig) using different methods, but as others have reported, soma sizes did not exhibit bimodalities near the fovea [46]. Soma size also did not distinguish between functional groups identified by chromatic tuning (Wilcoxon ranksum test, p > 0.1 for comparisons between L-M, S only, luminance only, and L-M/S cells).

### Spatial tuning of foveolar putative midget RGCs

Spatially patterned drifting gratings were used to probe the spatial frequency response of RGCs and to augment measurements of chromatic tuning (Fig 5). In M3, at the small field of view focused on the innermost RGCs, a control of spatially uniform mean luminance and drifting gratings of fourteen different spatial frequencies (561 nm, 6 Hz, 4–49 c/deg) were used to stimulate foveal cones while the 488 nm imaging light elicited fluorescence from the same GCaMP expressing RGCs shown in Fig 4B. At this temporal frequency, GCaMP could not track modulation, and responding cells exhibited a sustained increase in fluorescence in response to photoreceptor stimulation. A ΔF/F metric was calculated for each cell as the change in fluorescence under stimulation divided by the mean fluorescence in response to a grating of 17 c/deg (a spatial frequency with a robust response across the majority of the cells). Most, but not all of the putative midget RGCs at the foveal center in M3 (Fig 5), which were all identified according to their L-M or M-L cone opponency, showed two salient features when their spatial frequency response functions were measured: a strong low frequency cut, and a peak in the higher spatial frequencies (20–40 c/deg).

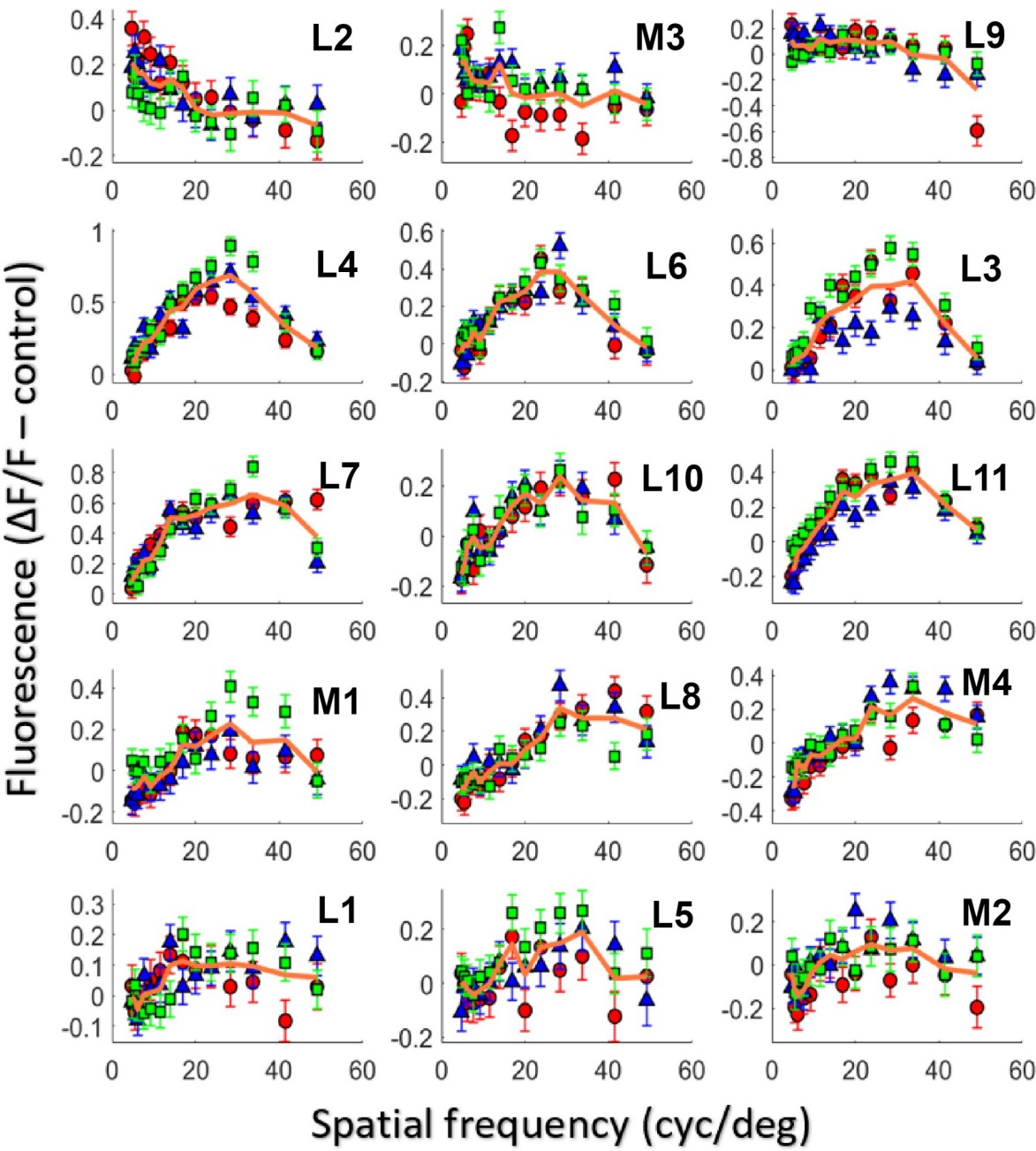

**Fig 5. Spatial frequency responses of putative midget cells.** Fifteen cells in M3 serving the central foveal cones were identified as L-M or M-L opponent (Fig 3D). Each cell is numbered L1-11 or M1-4 based on whether the suspected center cone was an L or M cone. The red points (circles, Week 1), blue points (triangles, Week 2), and green points (squares, Week 3) are measurements from three different experiments each one week apart. The solid orange line in each panel is the average response for that particular cell across all three experiments. The vertical axis is a response metric where the mean baseline fluorescence is subtracted from the fluorescence under stimulation and divided by the fluorescence under stimulation at the peak spatial frequency (see Methods). The error bars show the standard error at each data point, which represents the error in obtaining the signal and background needed to calculate the metric. Additional spatial frequency responses of cells from the large imaging FOV (S5 Fig) and non-midget RGCs (S6 Fig) can be found in the supplement.

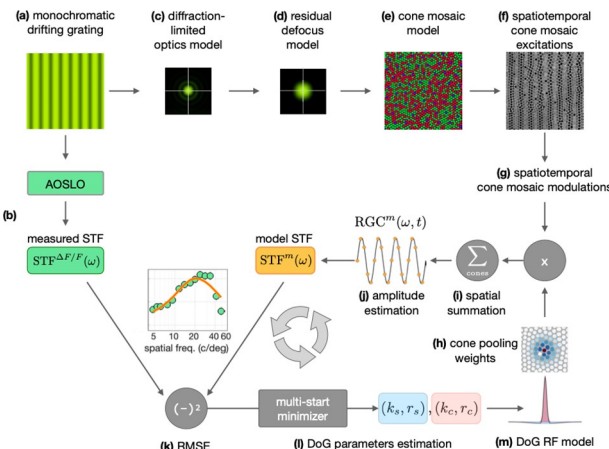

**Fig 6. Schematic overview of the ISETBio modeling approach.** To interpret the AOSLO-based STF measurements (Fig 5) in terms of the spatial pooling of cone signals to the measured RGCs we model the spatial, spectral and temporal aspects of the stimulus, the optics, the cone mosaic and the cone pooling within the center and surrounds mechanisms of an RGC using the ISETBio toolbox. **(a)** A frame of the drifting monochromatic grating stimulus which is projected through the AOSLO system to stimulate and measure the GCaMP fluorescence–based spatial transfer function (STF) of RGCs, $STF^{\Delta F/F}(\omega)$ **(b).** The drifting grating is modeled as a temporal sequence of spatial spectral radiance scenes. These scenes model the AOSLO display including a diffraction-limited optical system with a 6.7 mm pupil **(c)** combined with residual defocus in the eye's wave aberrations **(d).** The resulting sequence of retinal irradiance maps are spectrally integrated and spatially averaged within the apertures of cones in a model of the animal's cone mosaic **(e)**, resulting in a spatiotemporal sequence of cone excitations **(f).** Cone excitations are transformed to cone modulations by first subtracting and subsequently dividing the mean cone excitation level **(g).** Modulation signals from different cones are scaled by the spatial weights with which the RF center and the RF surround mechanism pool cone signals **(h).** The weighted cone signals are subsequently summed **(i)** to compute the temporal response of the model RGC. A sinusoid is fitted to the RGC temporal response **(j)** and the amplitude of the sinusoid is taken as the STF of the model RGC at the stimulus spatial frequency, $STF^m(\omega)$. The RMSE between $STF^m(\omega)$ and $STF^{\Delta F/F}(\omega)$ **(k)** is minimized using a multi-start solver which optimizes the center and surround peak sensitivity and radius parameters **(l)** of a difference of Gaussians (DoG) RF model **(m).** The model yields the best-fitting cone pooling weights for the RGC.

## Modeling of foveolar midget RGCs

Using the ISETBio-based approach described in the Methods and depicted schematically in Fig 6, we derived estimates of the spatial RF organization from the measured STF data under four different model scenarios involving combinations of residual instrument defocus (zero or some residual defocus) and RF center composition (single or multiple cone inputs). Model cone mosaics matched well in both density and cone size with the measured data in M3 along the horizontal and vertical meridians (S7 Fig). Modeling results from one RGC are depicted in Fig 7. Fig 7A shows results from the modeling scenario that assumes a single cone in the RF center and zero residual defocus. Note that this scenario predicts a diffuse surround and that the derived STF does not agree well with the measured STF, resulting in the largest RMSE across the four examined model scenarios. The STFs derived by the remaining three modeling scenarios match the measured STF approximately equally well even though the resulting cone pooling weights differ. The single cone RF center with 0.067 D residual defocus (Fig 7B) results in a more focal surround than would be true if there were no residual defocus. The multi-cone center RF with zero residual defocus (Fig 7C) results in a diffuse RF center and a surround that is marginally more diffuse than the center, which is not consistent with known anatomy of foveal midget RFs. Finally, the multi-cone center 0.067 D residual defocus scenario (Fig 7D) results in an RF center with heavy input from one cone and very weak input from nearby cones, and a focal surround. This scenario essentially collapses to the single cone center/0.067 D residual defocus scenario. In Fig 7E and 7F, the single and multi-cone RF center scenario

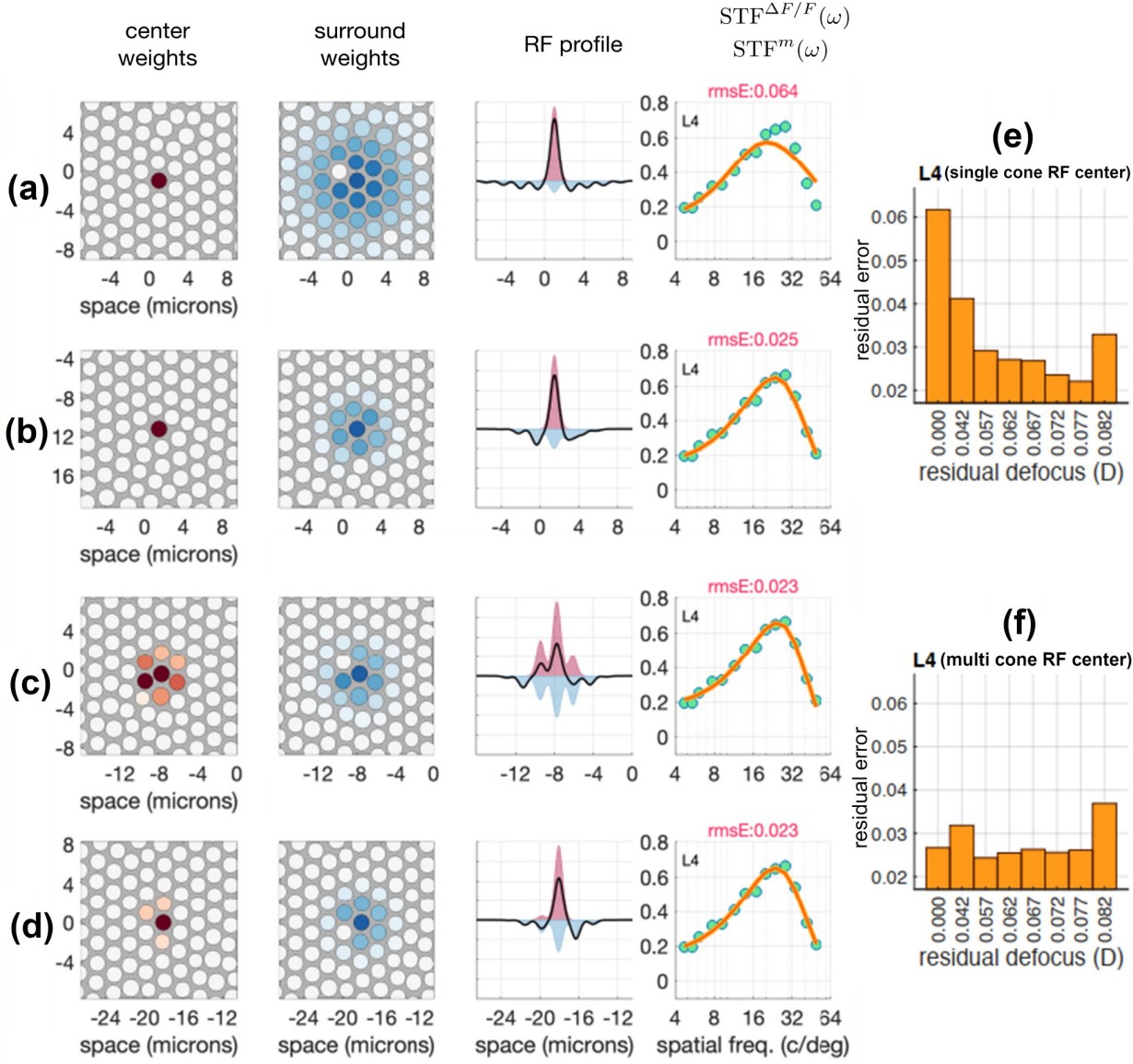

**Fig 7. Model interpretation of measured STF for a single RGC. a-b** show single cone center receptive field models with zero and 0.067 D residual defocus respectively. **c-d** show multi cone center receptive field models with zero and 0.067 D residual defocus respectively. In a–d, the first and second columns depict cone pooling weights for the center and surround mechanism, derived by fitting the model STF to the measured STF. The third column depicts the integrated (along the y-axis) center and surround sensitivity profiles, and the fourth column depicts the model-predicted STF (orange lines) together with the measured STF (green disks). **e-f** show single and multi-cone model residual errors for this particular cell with varying levels of residual defocus. The best residual defocus for individual cells in the single cone center receptive field modeled varies slightly, so 0.067 D was chosen as the value optimizing models across all cells measured.

performances are depicted as a function of assumed residual defocus. Note that for the particular cell shown, the minimal error in the single cone center scenario was at 0.077 D of residual defocus, and that the multi-cone scenario never performs better than the best single cone scenario at all residual defocus values modeled. Across all cells examined, the best performance for the single cone RF center scenario was obtained for values of residual defocus between 0 D and 0.082 D with a mean value of 0.067 D. The comparison shown for one cell in Fig 7E and 7F is provided for all cells modeled in S9 Fig. In addition, a cross-validation analysis (S8 Fig)

similarly suggests that across all cells, the data do not reject the single cone RF center/0.067 D model relative to the multi-cone center models. Therefore, for the remainder of this paper, we use the single cone RF center/0.067 D residual defocus scenario as the vehicle for interpreting the measured STFs.

Fig 8 depicts the results of the single-cone RF center/0.067 D scenario for four cells from the center of the fovea of M3. These cells were chosen to showcase the model's fitting to slightly different STF shapes and are not necessarily special in any way. Full results of the model fitting to all 15 cells, including fits for all four scenarios examined, can be found in S10 Fig.

The ISETBio model expresses the RFs directly in terms of the weights with which they pool signals from cones in the mosaic, and as such is appropriate for comparison with anatomical measurements and with *in vitro* physiological data where the optics of the eye are not in the stimulus light path. In order to compare our data to other *in vivo* data collected in macaque [e.g. 9], where the eye's optics form a blurred image of the visual stimulus, we predicted *in vivo* STFs using the cone pooling weights obtained from the single cone RF center/0.067 D residual defocus scenario. To do so, we swapped the AOSLO optics with the animal's (M3) own wave-front aberration optics as measured *in vivo* and assumed a pupil diameter of 2.5 mm. These physiological optics STFs were computed for 100% contrast achromatic sinusoidal gratings, which matched in their spatial and temporal properties the monochromatic gratings used to measure the STFs in the AOSLO. Results of this analysis are displayed in Fig 9, with three example cells shown in Fig 9A. Here, gray disks and black lines depict the AOSLO STF data and corresponding fit of the model for the AOSLO measurement conditions (single cone RF center/0.067 D residual defocus). Red disks depict the STFs computed using the model combined with M3's optics, and red lines depict a simple DoG model fit to the computed STFs, as is typically done in analysis of *in vivo* data. Note that the STFs predicted under physiological optics differ substantially from the STFs obtained under near diffraction-limited optics: their magnitude is lower, and their shape is less bandpass. They have weaker attenuation at low frequencies and peak at lower spatial frequencies. This is due to the differential effect of the optical modulation transfer function (MTF) to the STF of the center and surround mechanism, as illustrated in Fig 9B. The left panel depicts the STFs of the center (pink) and surround (blue) mechanism of a model RGC under diffraction-limited optics, and the gray disks represent the corresponding overall STF of the cell. The middle panels depict MTFs for three hypothetical physiological optical systems with increasing amounts of Gaussian blur. The physiological STFs that would be obtained under the three MTF conditions are depicted in the right panels of Fig 9B. Note that as the physiological optics become more blurred, the mapped STF becomes less bandpass and its peak shifts towards lower spatial frequencies. This occurs because the difference between center STF and surround STF decreases as the optics become more blurred. Also note the overall decrease in the magnitude of the STF predicted under physiological optics conditions.

To compare our data with published results on midget RGC STFs obtained *in vivo* from outside the central fovea, we fit the DoG model to the synthesized STFs obtained under the animal's optics and we contrasted the DoG parameters derived using the procedure illustrated in Fig 9A to those reported by Croner & Kaplan [9]. Fig 10 shows the characteristic radii of centers and surrounds of data from M3, as predicted with physiological optics, along with data from Croner & Kaplan. Cone characteristic radii as measured in M3 (blue circles) along with those obtained in *Macaca nemestrina* from Packer *et al.* [99] (blue dashed line) are also shown. Using eccentricity dependent optics derived from off-axis wavefront aberration measurements taken by Jaeken and Artal [100] from human subjects, the anatomical cone apertures from Packer *et al.* were transformed into their visual space counterparts. This transformation was accomplished by convolving the cone aperture by the point spread function at the

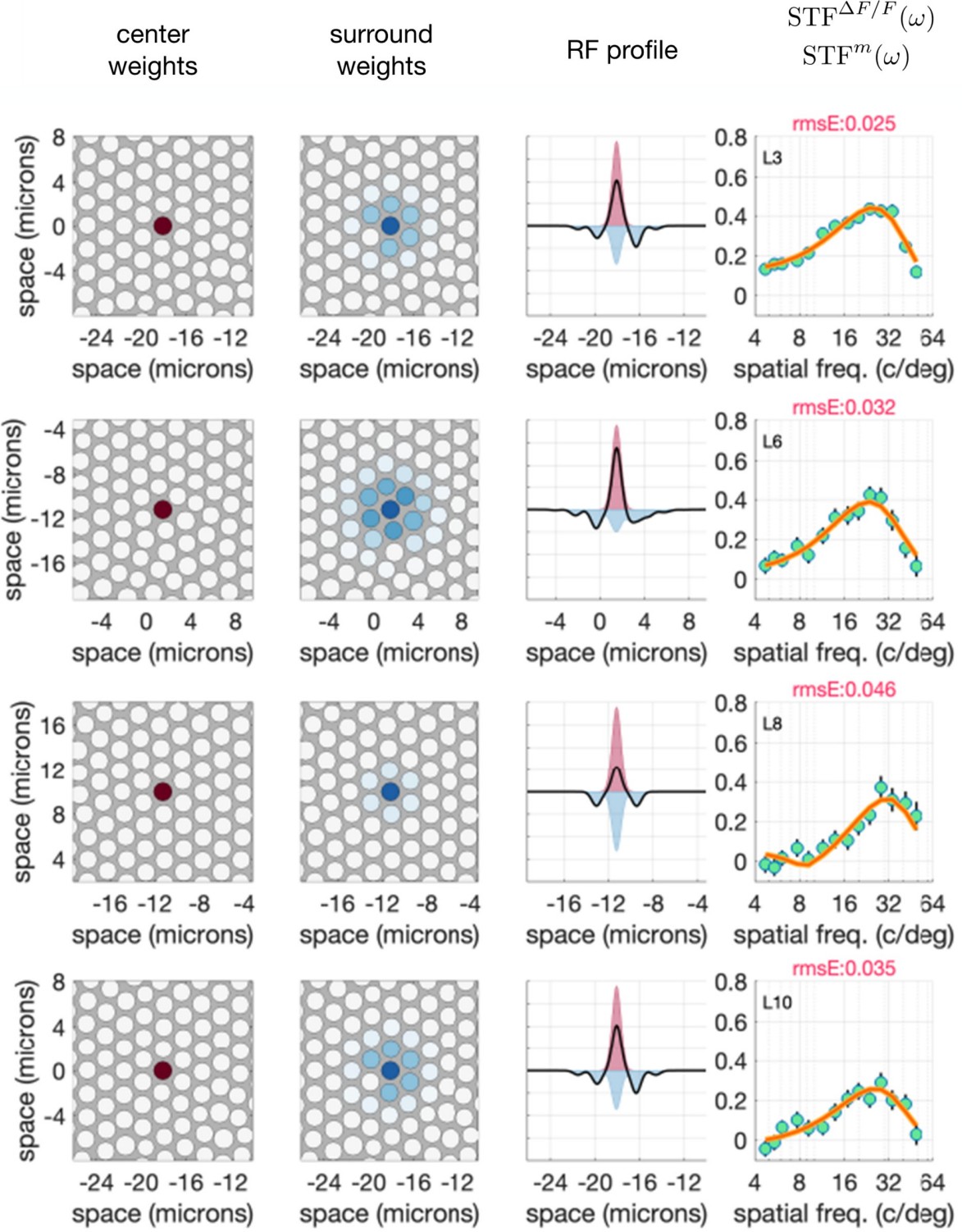

**Fig 8. ISETBio model receptive field estimates for four additional RGCs.** Each row shows the model fits to a different cell's STF using the single cone center/0.067D residual defocus model.

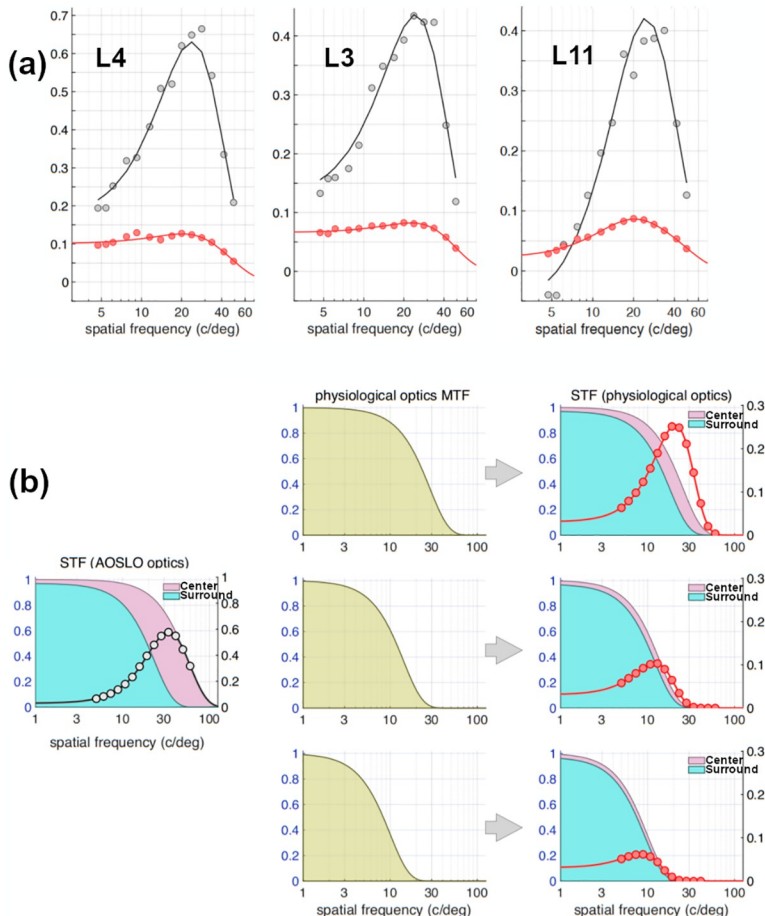

**Fig 9. Effect of physiological optics on RGC STFs. a.** Gray disks depict STF data measured using the AOSLO for three RGCs, and the black lines show the STFs from the ISETBio model fit to these cells. The cone weights from the model fits are used to predict the STF that would be measured under the animal's own physiological optics (as characterized by wavefront-aberration measurements taken during the experiment) with a 2.5 mm pupil. These model-predicted STFs are depicted by the red disks. A simple DoG model is then fitted to the predicted STF data (red line) for comparison to measurements obtained in traditional *in vivo* neurophysiological experiments (e.g. Croner & Kaplan [9]). **b. Demonstration of the effect of physiological optics on the STF.** The left panel depicts the STFs of the RF center and the RF surround of a model RGC as they would be measured using diffraction-limited optics (pink and blue, respectively), whereas the composite STF is depicted by the gray disks. The middle panels depict the MTFs of three hypothetical physiological optical systems with progressively larger Gaussian point spread functions. The corresponding STFs that would be measured under these physiological optical systems are depicted in the right panels. Note the difference in scale between left y-axis (for the center and surround STFs) and right y-axis (for composite STFs).

corresponding eccentricity, and fitting the result with a two-dimensional Gaussian to derive the characteristic radius in visual space. The result of this computation is shown by the blue cyan line in the right panel. The midget RGC center and surround characteristic radii from M3 lie along a reasonable extrapolation from the Croner & Kaplan data, once eccentricity-varying optics and cone aperture size are accounted for. This supports the hypothesis that the midget pathway is specialized to preserve information at the level of the cone photoreceptor mosaic, and that midget RGCs at the foveola transmit high acuity chromatic information from retinal images.

Fig 11A depicts population data for $R_c/R_s$, the ratio of center to surround characteristic radii, which typically shows a smaller center size compared to the surround in the Gaussian

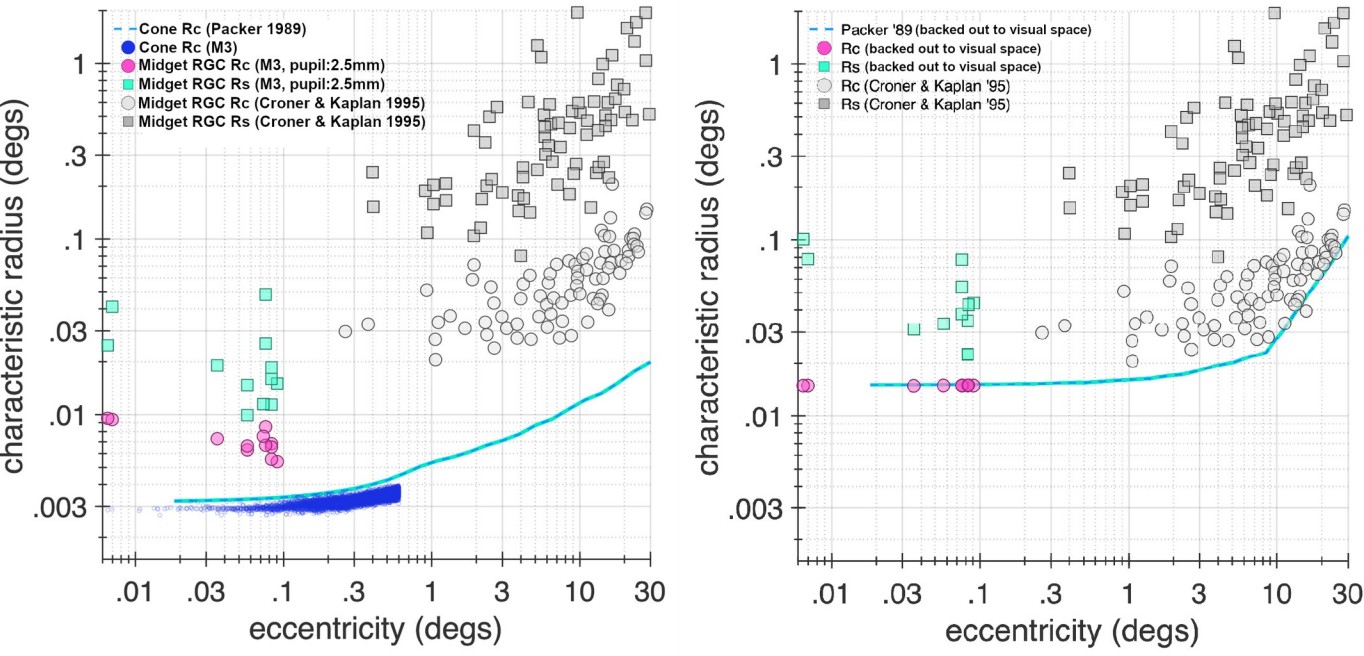

**Fig 10. Comparison to extra-foveal *in vivo* midget RGC data.** Data from M3 model fits are compared with extra-foveal data from Croner & Kaplan [9]. At left, gray circles and squares respectively show center and surround characteristic radii from Croner & Kaplan as a function of eccentricity, while pink circles and green squares show characteristic radii of centers and surrounds from M3 predicted using the ISETBio model with physiological optics accounted for. Light blue dashed line shows cone characteristic radii data from Packer *et al.* [99], while blue dots represent cone characteristic radii measured in foveal cones of M3. At right, the cone apertures of Packer *et al.* were transformed to their visual space counterparts using eccentricity dependent optics derived by off-axis wavefront measurements from human subjects from Jaeken & Artal [100] (blue line). The center and surround characteristic radii (pink circles and green squares) from M3 are also projected into visual space and lie along a natural projection of the Croner & Kaplan data.

model and can vary across RGC types. The data of Croner & Kaplan [9], which were obtained from a wide range of eccentricities, are depicted in gray, whereas data from our foveal RGCs are shown in blue (AOSLO optics) and yellow (M3 as predicted for physiological optics). Note that our population of foveal RGCs has a higher mean $R_c/R_s$ ratio (0.32 under AOSLO, 0.45 under M3 optics) than the population of Croner & Kaplan (0.15), which is not surprising since foveal RGCs are expected to have smaller surrounds than peripheral cells [6, 8, 20].

Fig 11B depicts population data for $K_s/K_c$, the ratio of surround to center peak sensitivity, a measure of the surround compared to the center strength weighting, where typically it has been found that the surrounds are much weaker than the centers when expressed in terms of peak sensitivity [9]. Again, there is a difference between our foveal cells with mean $K_s/K_c$ values of 0.28 under AOSLO optics and 0.11 under M3's optics, whereas the Croner & Kaplan data have a mean of 0.0094. To the extent that the midget RGC data from M3 and the Croner & Kaplan study [9] are comparable, this suggests that the surround strength expressed as a peak sensitivity ratio is stronger at the foveola than at more eccentric locations, supported by the general trend of the $K_s/K_c$ ratio as a function of eccentricity in both the Croner & Kaplan and M3 data.

Fig 11C provides the surround/center integrated sensitivity ratio, which is essentially the ratio of the total sensitivities (in a volumetric sense) of surrounds and centers. This does not vary with eccentricity in the Croner & Kaplan data [9], as the variation in $R_c/R_s$ and the

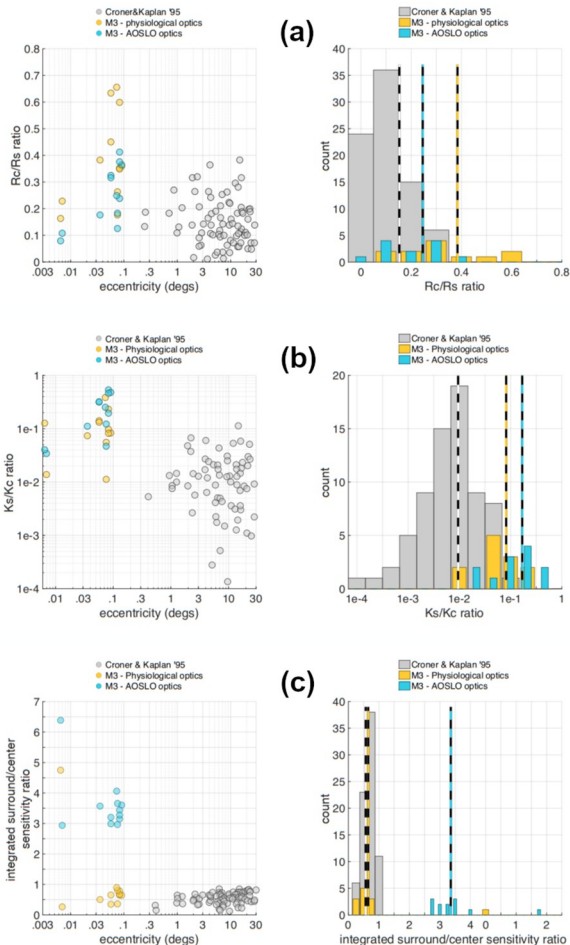

**Fig 11. Relationship between our findings in foveal midget RGCs and findings in midget RGCs from earlier studies.** Data from Croner & Kaplan [9] is shown in gray, data from M3 using the ISETBio model and physiological optics is shown in yellow, and data from M3 using the AOSLO optics is shown in blue. **a.** Analysis of Rc/Rs ratio. Left panel: relationship between the Rc/Rs ratio and retinal eccentricity. Right panel: distribution of Rc/Rs. The mean Rc/Rs ratio in the Croner & Kaplan data is 0.15, vs. 0.45 in our foveal RGCs under physiological optics, and 0.32 in our foveal RGCs under AOSLO optics. **b.** Analysis of Ks/Kc ratio. The mean Ks/Kc ratio in the Croner & Kaplan data is 9.4e-3, vs. 1.1e-1 in our foveal RGCs under physiological optics, and 0.28 in our foveal RGCs under AOSLO optics. **c.** Analysis of integrated surround-to-center ratio, $(Ks/Kc) \times (Rs/Rc)^2$. The mean integrated surround-to-center ratio in the Croner & Kaplan data is 0.54, vs. 0.71 in our foveal RGCs under physiological optics, and 3.37 in our foveal RGCs under AOSLO optics.

variation of $K_s/K_c$ counteract each other such that the integrated sensitivity ratio is relatively constant across midget cells. Consistent with this, the integrated sensitivity ratio for M3 predicted for physiological optics conditions is similar to that found in Croner & Kaplan's data (means of 0.71 and 0.54 respectively). Note also the large effect that the optics have on this value, which rises to 3.37 under the measured AOSLO conditions. Overall, this population analysis shows that our foveal RGCs have a large degree of surround antagonism when the receptive field is expressed directly in terms of how the center and surround draw on the cones in the mosaic, but that this antagonism is greatly attenuated under physiological optics viewing conditions, resulting in center/surround structure in our data that is consistent with what has been observed in previous midget RGC studies across a wide range of eccentricities.

## Discussion

In this study, we repeatedly imaged the same RGCs serving the foveal center *in vivo*, in three animals, returning to the same retinal locations each time. Spatially uniform chromatic visual stimuli enabled the identification of L-M and M-L opponent cells which were further interrogated in one animal with drifting gratings of various spatial frequencies. Detailed modeling revealed how the receptive fields of these RGCs draw on the cones of the retinal mosaic. The L vs. M opponency and high spatial frequency responses of these cells identifies them as candidate midget RGCs. Our modelling is consistent with the hypothesis that these cells have single cone center inputs and are specialized to preserve the resolution of the cone photoreceptor mosaic in the signals they send to the visual cortex.

### Measuring the precise location of cells from the foveal center

In our data, the foveal avascular zone (FAZ) was used as the 'center' measurement of the fovea, as the imaging paradigm precluded using the location of maximum photoreceptor density for all sessions. The preferred retinal locus of fixation (PRL) is also of interest but could not be determined using our method, since the animals are anesthetized during the experiments and cannot freely fixate. The center of the ganglion cell ring surrounding the fovea is also sometimes used to define the center of the fovea, but this measure is highly correlated with the center of the FAZ [11, 101]. A more sophisticated measure of the 'physiological center' can be identified using white noise stimuli as in McGregor *et al.* [16], but those stimuli were not used in this study. Others have characterized the precise nature of the relationship between these various measures [14], and as noted in the results, our measurements of the difference between the center of the FAZ and the location of maximum cone density in M3 are in agreement with that literature. In any case, the difference in identification of the 'foveal center' is not large enough under our method to affect the conclusion that, using the mapping of McGregor *et al.* [16], the receptive fields we examined are located near the very center of the foveola.

### Comparison of methodology used in this study with traditional electrophysiology

Functional measurements of RGCs made in the past often involved flashed stimuli and marking approximate estimates of soma location on fundus images of the retina [e.g. 9, 102], a method that leads to uncertainty about precise locations. In contrast, our imaging method captures photoreceptors, vasculature, and RGC somas in the same field of view, allowing more accurate measurements of receptive field eccentricity and soma position. One shortcoming of *in vivo* functional imaging compared to traditional electrophysiology is that the excitation and imaging sources can interfere with stimulation or fluorescence detection. However, unlike in the periphery, foveal RGCs are displaced laterally from their photoreceptor inputs [1, 11, 15, 98]. Thus, our paradigm avoids this pitfall, and we can reliably image foveal RGCs without the 488 nm excitation light stimulating foveal cones (Fig 2C) or the photoreceptor stimuli interfering with the RGC GCaMP fluorescence imaging. As mentioned in the Methods, our technique affords the advantage of being able to record from hundreds of cells simultaneously, a feat matched by multi-electrode arrays [e.g. 35, 39, 49, 55, 103–105] but without the same spatial control or ability to view the cells during recording. However, when imaging hundreds of cells from a wide field of view in our technique, there is the possibility of optical crosstalk from cell responses originating from different depths in the ganglion cell layer. Currently, the only way to get around this in our existing system is to use smaller fields of view focused on the foveal slope where ganglion cells are in a monolayer. In the future, methods that improve the axial

resolution of *in vivo* imaging, such as combining adaptive optics with optical coherence tomography [e.g. 46, 106–108], will be required to achieve the same level of single-cell specificity that that is achievable with current electrophysiology techniques.

## S-cone mediated color vision in the fovea

As shown in Figs 3 and 4, cells with either S cone input without L vs. M cone opponency (pure) or S cone input on top of L vs. M opponency (mixed) were observed in each animal. 40% of responsive cells exhibited pure S responses and 7% mixed in M1, 9% pure and 14% mixed in M2, 16% pure and 3% mixed in the near fovea of M3 (large imaging FOV), while no consistent S cone responses were recorded from the centermost foveolar cells (small imaging FOV) in M3. Other than in M1 where the sample size was only 15 cells, these numbers are consistent with the expected S cone density of approximately 10% across primate fovea [109] and the known reduction of S cone density at the very center of the primate fovea [8, 15]. Others have reported subtypes of midget RGCs with S cone input or even that midget RGCs may indiscriminately contact S cones along with L and M cones in the surround [4, 17, 48, 110, 111]. Our current data set has no repeatable S cone responses for RGCs at the innermost foveal edge in M3, where RGCs are optically well-isolated, so we cannot rule out that the mixed cells we measure are partially or fully the product of optical crosstalk resulting from the 25–30μm axial resolution of the AOSLO instrument.

## Putative parasol cells at the fovea

As shown in Figs 3 and 4, cells with no chromatic responses but only a L+M or -L-M luminance response were recorded from each of the three animals: 7% of responsive cells in M1, 4% in M2, 15% in the entirety of M3 and 20% of the centermost foveal cells in M3, consistent with expected proportions of parasol ganglion cells, which are known to mediate achromatic vision [20]. Simple modeling based on hexagonal packing of cones and random wiring of 6 cones to midget surrounds suggests that approximately 9% of midget RGCs could be dominated by a single cone type and appear achromatic (S11 Fig). It is probable then that some of these achromatic cells are midget RGCs but that most of them are candidate parasol cells, though at this time we cannot make that distinction for an individual cell. The spatial frequency response properties of these cells were even more varied than the chromatic opponent cells (S6 Fig), suggesting that they might be responsible for a wide range of achromatic or even temporal vision properties that complement the midget system. As noted in the results, we have not found significant differences between soma sizes of putative foveal parasol and midget RGCs, so additional anatomical information such as histological studies may be required in the future to confirm functional designations.

## The role of midget RGCs in foveal vision

Among the putative midget RGCs from M3 (Figs 5 and 7–9), there was some variability among the responses to spatial frequencies in that a subset of cells possessed lower spatial frequency response peaks (0–20 c/deg) compared to the peak spatial frequencies of the majority of the cells (20–40 c/deg), though both of these groups of cells were well fit by the single-cone RF center/0.067 D residual defocus model scenario, suggesting that the change in their peak spatial frequency is related to the surround size and/or ratio of center and surround strengths, as opposed to the spatial extent of the center. Previous studies examining parvocellular neurons in the lateral geniculate nucleus (LGN) serving the central five degrees of retina found an average of 4.57±2.75 c/deg as the peak spatial frequency response [62], and other studies in the retina found parvocellular cells in the central five degrees to have a peak spatial frequency

response around 3 c/deg [9]. However, these previous studies relied on presenting stimuli through the animals' optics, whereas the adaptive optics method we use approaches near diffraction-limited stimulus presentation with some possible residual defocus. Indeed, as shown in Figs 9 and 10, when the animal's optics are added back into a simulation of our data from M3, we recover the classically measured bandpass shape and lower spatial frequency peaks that are expected of parvocellular neurons (midget RGCs). We thus measure putative midget RGCs not only closer to the foveal center and with greater precision than previously achievable, but also with diminished optical blur in the *in vivo* retina compared to previous studies, with the exception of studies that used interferometric stimuli to bypass the optics altogether [e.g. 10]. We thus posit that the parameters of spatial DoG models fit to data collected through the optics do not directly represent the *in situ* characteristics of RGCs, and that future *in vivo* measurements and models need to take into account the effects of diffractive and optical blurring on the measured spatial transfer functions.

In all 15 L-M and M-L cone opponent RGCs serving the central foveola, we could not reject the hypothesis that the receptive fields were best described by a single cone center with some residual instrumental defocus (0.067 D). There were some cells where the single cone center/ 0.067 D scenario was clearly the best fit, but for many cells several of the scenarios performed similarly in terms of RMSE—full details for all scenarios can be found in S10 Fig. We posit that since the fovea is specialized for high acuity vision, it must be capable of extracting high spatial frequency information from natural scenes which are known to follow a $1/f^2$ law [112] and are dominated by low frequency information. Our modeling that adds the optics of M3 back into the predictions (Fig 9) suggests that natural ocular aberrations reduce the peak spatial frequency obtained under AOSLO conditions by a factor of three or more, which means that high spatial frequency information is greatly reduced by ocular transmission. Thus, to extract meaningful high spatial frequency information from natural scenes through the eye's native optics, detectors that preserve information at the resolution of the cone photoreceptor mosaic are required. It seems highly probable that the putative midget RGCs shown in Figs 5 and 8 are responsible for this resolution, as they have responses consistent with single cone receptive field centers and most have spatial transfer functions that peak well into the 20–40 c/deg range when measured with respect to a retinal image under AOSLO conditions. McMahon *et al.* [10] measured the spatial frequency responses of parvocellular LGN cell centers near the foveola using interference fringes, which are not subject to blur by diffraction. They found that cells still had detectable responses to spatial frequencies above 100 c/deg, with some responses that were complex and inconsistent with a simple difference of Gaussians model. Because the AO-corrected stimuli used in the present study were subject to blur by diffraction, we were unable to explore responses at the high spatial frequencies where such effects were observed.

Putative midget RGCs with the lowest spatial frequency peaks comprised only 2/15 cells (cells L2 and M3) compared to ones with higher spatial frequency peaks (13/15 cells). The two cells with lowest spatial frequency peaks had stronger surround weighting compared to the centers (higher $K_s/K_c$ ratio) coupled with surrounds that were almost spatially coexistent with the centers. These two cells are inconsistent with previous data finding the center responses of midget RGCs to be some 10x-1000x stronger than those of the surround and finding centers to be much smaller in spatial extent than surrounds [9]. The cells with higher spatial frequency peaks had center and surround strength ratios closer to previous data, though all 15 cells had higher surround/center integrated sensitivity ratios than previous results such as those reported by Croner & Kaplan [9]. These differences, however, are explainable by the blurring effect of the optics as presented in Figs 9–11. Further study of low-frequency L-M and M-L cone opponent cells will confirm if they exhibit any other physiological differences or if they belong to one or more of several midget subtypes as others have postulated to exist [48, 110].

There is also evidence that parasol ganglion cells can respond weakly to L vs. M modulation [113], but they are not expected to have single cone centers from anatomy [6, 20, 27, 114], so it is unlikely that any of the 15 cells studied here are parasols.

## Modeling foveolar RGCs

In our modeling, we could not reject the hypothesis that the 15 cells studied from the center of the fovea in M3 had single cone centers with some residual optical blur, beyond diffraction, present in the measurements. We think it likely that there was some residual optical blur, and certainly cannot reject this idea. However, we note that model scenarios that did not restrict to a single-cone center also describe the data, and that if no residual blur is assumed then the single-cone center model can be rejected for some cells. There are two reasons it is difficult to differentiate between single and multi-cone scenarios from our data. First, due to limitations imposed by the pixel density of the raster scan, our measurements did not extend to high spatial frequencies (>50 c/deg) where the different scenarios would make more clearly different predictions. Second, there was inter-session variability in the measured STFs (Fig 5), which decreases the power of the data. In future experiments, we plan to (i) extend the range of spatial frequencies examined and (ii) test multiple grating orientations. Despite these limitations, as noted in S10 Fig, the multi-cone RF scenario with residual defocus often produces a RF strikingly similar to the single-cone RF scenario with residual defocus, with only a minimal amount of cone coupling, and the classical understanding of midget RGCs that postulates single-cone centers is consistent with our data.

A notable observation of the STFs in our population of foveal RGCs (Fig 8) is the large amount of low-spatial frequency attenuation. Such observations are not typical of midget ganglion cells reported in other *in vivo* studies [9]. The question then arises as to whether these most central foveal cells have distinctively different RF organization from the less-foveal cells reported in other studies, or whether the underlying RF organization is similar, but the measured data differ between the two types of studies simply because of differences in the optical systems involved. When the animal's optics are added back in (Figs 9–11), the STFs have a clear shift of peak sensitivity towards lower spatial frequencies, approaching those of previous *in vivo* studies but not exactly replicating them, suggesting that the optics account for most of the observed differences with the possibility of modest foveal specialization or instrument differences accounting for the rest.

The modeling used in this study has the capability to add a subject's optical point spread function (here through the measured wavefront aberrations from M3) to predict responses to stimuli presented through the natural (non-AO-corrected) optics. The robust nature of the entire simulation allows for any cone density profile or PSF to be substituted in the model so that the simulation can be generalized to population data or be tailored to individual subjects, as it has been here for M3.

## Conclusions

In this study, measurements of chromatic and spatial properties were repeated for hundreds of foveal retinal ganglion cells across three macaque animals. Cells with S cone responses were found in proportions consistent with known S cone density at the fovea, as were achromatic cells found in proportions consistent with known parasol RGC densities. Putative midget RGCs at the center of the foveola were identified by their unique L vs. M cone opponency and had spatial responses consistent with single cone centers. Two out of fifteen RGCs with L vs. M opponency were identified as being inconsistent with the established view of midget cells, in that although their responses were consistent with single cone centers, they have extremely

strong surround opponency that is almost spatially coextensive with the centers. In general, the putative midget cells were consistent with measured responses from outside the fovea when the blurring effect of diffraction and the animals' optics was taken into account. Spatial frequency response measurements made using adaptive optics are closer to the *in situ* properties of RGCs than other *in vivo* methods as shown by our modeling, and future models of retinal neurons will need to take into account the effect of optical blurring on spatial transfer functions to connect the *in situ* properties to the cell's function for natural viewing.

## Supporting information

**S1 Fig. AOSLO system diagram.** The adaptive optics scanning light ophthalmoscope used in this study consists of three main arms or subsystems. First, the Source arm, contains the 488 nm laser used to excite GCaMP fluorescence, the 561 nm laser used as visual stimulation as in the drifting gratings, the 796 superluminescent diode used to image the cone photoreceptors, and the 843 nm laser diode used for wavefront sensing and correction. Each source is co-aligned and combined with dichroic mirrors so that at the output of the arm they exit coaxial with each other along the same beam path into the system. 90% of the source power is lost at a 90/10 beamsplitter such that 10% of the source power enters the Sample arm of the system—this is done so that on the back-pass, 90% of the signal is transmitted to the detectors while only 10% of the signal is lost. The Sample arm is composed of various optical telescopes (whose function is to translate the beam axially and change the magnification to match the pupil sizes of the various active elements) and four important planes conjugate to the pupil. The first two conjugate planes are the vertical (VS) and horizontal (HS) scanners which raster scan the beam across a rectangular field of view. The third conjugate plane is the deformable mirror (DM) which changes shape to correct the measured wavefront aberrations of the animal's eye to provide near-diffraction limited imaging of the retina. The fourth conjugate plane is the pupil of the animal's eye, where all source light enters and from which all detected light emanates. In the Detection arm, two photomultiplier tubes (PMT) detect visible fluorescence (VIS) from the emitted GCaMP signal and infrared (REF) reflected light from the cone photoreceptors. A Shack-Hartmann wavefront sensor (SHWS) collectes infrared light from the 843 nm source and measures the wavefront aberrations of the animal's eye, sending that information to the deformable mirror in a closed loop. Finally, a Maxwellian View subsystem is used to inject the LED stimulation lights into the system near the pupil plane (bypassing both scanners and the deformable mirror). The subsystem contains LEDs at 420 nm, 470 nm, 530 nm, and 660 nm, although the 470 nm LED was not used in this study. Just as in the Source arm, the LEDs are co-aligned and combined using dichroic mirrors so they follow the same beam path. The LEDs pass through a spatial filter which removes any spatial inhomogeneities and then through a field stop which restricts the beam sizes on the retina to a circular subsection of the fovea about 1.3 degrees in diameter. The LEDs are reflected into the system via a pellicle beamsplitter which has 88% transmission (throws away 88% of the LED source power, but allows 88% of the signal to pass on the way to the detectors).
(PDF)

**S2 Fig. Examples of raw fluorescence time courses of putative midget cells to various stimuli.** Six of the putative midget cells identified from the foveola of animal M3 are shown in detail. For each cell, the fluorescence time course is shown for that cell for six different stimuli: the L-isolating 0.15 Hz flicker, the M-isolating 0.15 Hz flicker, the S-isolating 0.15 Hz flicker, the 9.2 cyc/deg 6 Hz drifting grating, the 28.3 cyc/deg 6 Hz drifting grating, and the 49.1 cyc/deg 6 Hz drifting grating. For each stimulus, there are three fluorescence traces corresponding to experiment one (blue), experiment two (orange), and experiment three (yellow), which all

occurred approximately a week apart. There is a fourth fluorescence trace (black) that is the average over the three experiments shown. Each fluorescence time course was normalized to the peak response for that cell, and a moving window (MATLAB movmean()) was used to smooth each fluorescence time course with a width of three seconds for the L/M/S isolating stimuli and a width of 5 seconds for the drifting grating stimuli. Note that as described in the manuscript, the L and M isolating stimuli are characterized by a modulation of the fluorescence near the stimulus frequency of 0.15 Hz. As these six cells were all putative midgets, note that the S isolating responses do not have the same modulation frequency, and are mainly affected by the respiration rate of the animal which causes small residual motion of the eye (approximately 0.28 Hz in M3). Note that as described in the manuscript, the drifting grating responses are characterized by a sustained increase in the mean fluorescence as opposed to a modulation, since the drifting speed exceeded the GCaMP6 temporal sensitivity.
(PDF)

**S3 Fig. Variability of responses to chromatic and achromatic flicker.** Plots for the small FOV from M3 show the response variability to the L isolating, M isolating, S isolating, and Luminance stimuli. Note that these are boxplots showing the median, 25–75% interquartile range, and lowest and highest values, but there were only three measurements. Thus, the median value and the two ends of the boxplots represent the three measured values for each cell. For the large FOV from M3 and M2, there are many more cells, so the standard error divided by the mean SNR is plotted as a histogram for all cells that were responsive to each individual stimulus. No standard error can be plotted for M1, as there was only one experiment in that animal.
(PDF)

**S4 Fig. Soma sizes of RGCs for animals M2 and M3 across chromatic functional groups.** Observer 1 used the open source software GIMP to segment RGCs in fluorescence images from M2 (large FOV), M3 (large FOV) and M3 (small FOV). The ellipse tool was used to segment the rough boundary of individual RGCs. Observer 2 used the open source software ImageJ to segment the same RGCs in fluorescence images from M2 and M3 using a hand tracing tool to trace the observable edges of each cell's fluorescence. Under both methods, the area for each cell was computed in terms of pixels$^2$ and then converted to μm$^2$ using the following formula: $\frac{\mu m}{pixel} = 291.2 \frac{\mu m}{deg} * \frac{axial\ length}{24.2\ mm} * \frac{FOV\ width}{496\ pixels}$, where 291.2 μm/deg is the human model eye visual angle to retinal extent conversion, 24.2 mm is the human model eye axial length, 496 pixels is the width of the imaging PMT used in the AOSLO system, axial length is the animal's axial length in mm, and FOV width is the width in degrees of the FOV used. The axial length of animal M2 is 17.2 mm, and the FOV width used was 3.64 deg. The axial length of animal M3 is 16.56 mm, and the FOV widths used were 3.69 deg (large FOV) and 2.54 deg (small FOV). Cells were compared across functional groups identified as L-M/M-L chromatic opponent (L-M), S only responding (S), Luminance only or achromatic (LUM), and mixed L-M/S responses (L-M/S). In **(a)**, the soma areas in μm$^2$ for the four functional groups in M2 are listed as the mean and standard deviation of each group across both observers. In **(b)**, the same comparisons are made for the four functional groups in M3 at the large FOV. In **(c)**, comparisons for the two functional groups found in M3 at the inner edge of the foveal slope (small FOV) are made across both observers. In, **(d)**, summary tables show the p-scores from a Mann-Whitney U test (MATLAB function ranksum(x,y)) comparing the distributions of soma areas across functional groups as measured by both observers separately. All p-values were greater than 0.1, except for Observer 2's comparison of the L-M group to the S only group in M2 which had p = 0.0118. Based on these results, we cannot reject the hypothesis that the

distributions of soma sizes for these functional groups are roughly the same across the two animals measured at the range of eccentricities at which we imaged cell somas.
(PDF)

**S5 Fig. Spatial response of RGCs from M3 in the large FOV condition.** Some example responses of L-M/M-L chromatic opponent RGCs from M3 at the large FOV, where each plot is the response of a different cell to drifting gratings (6 Hz) of spatial frequencies from 2–34 c/deg. These cells were chosen to showcase different cell responses and are not necessarily special or representative of the population. Purple and Orange curves show two separate experiments that were averaged together (Gray curve). The Blue curves are simple difference of Gaussians fits (not the full ISETBio modeling, for reasons explained below) to each average. Each plot title includes the cell's unique label "cN", and five parameters—the center strength $K_c$, the center radius $r_c$, the surround strength $K_s$, the surround radius $r_s$, and a goodness of fit value for the difference of Gaussians. As can be easily seen, many of the cells do not exhibit high spatial frequency falloff at the range of spatial frequencies measured, and so the difference of Gaussians either fits poorly or produces fit parameters that are highly irregular or suspect such as center sizes much smaller than the size of a single cone. For this reason, these data were not incorporated into the modelling in the main body, as higher spatial frequency responses were needed to properly fit many of these cell responses. There were 48 such L-M/M-L chromatic opponent cells in the large FOV in M3 out of 145 measured cells. In M2, there was only one experiment measuring response to spatial frequency and it suffered from the same lack of high spatial frequency measurements as the data shown here. Full data from both animals can be shared upon request.
(PDF)

**S6 Fig. Spatial response of achromatic RGCs from M3 at the center of the foveola.** Each graph represents a single cell identified as responding only to achromatic stimuli from the center of the foveola of M3. The measured responses were to drifting gratings (6 Hz) of spatial frequencies from 4–49 c/deg. Purple, Orange, and Green lines represent the cell's response in three different experiments. The Gray line is the average of the three experiments for each cell, and the Blue line is a simple difference of Gaussians fit to the average (not the full ISETBio model, see S5 Fig). Each plot title includes the cell's unique label "cN", and five parameters—the center strength $K_c$, the center radius $r_c$, the surround strength $K_s$, the surround radius $r_s$, and a goodness of fit value for the difference of Gaussians. Compared to the 15 L-M and M-L cone opponent cells from this same location, these 7 achromatic cells were had noisier responses across experiments and more varied response characteristics. Though most of these cells are likely to be parasol RGCs, some could be achromatic midget RGCs or other rarer achromatic RGC types. Due to this uncertainty, these data were not included in the modelling for putative midget RGCs presented in the main manuscript. There were also achromatic cells identified in M2 and in M3 at the large FOV condition, but those data suffer from decreased resolution and a decreased range of spatial frequencies presented to the cells. Full data from all animals can be shared upon request.
(PDF)

**S7 Fig. Cone mosaic modelling fits compared to data from M3.** Images showing the performance of the ISETBio cone mosaic generator compared to the actual data from M3. At top, the model trichromatic photoreceptor mosaic generated by ISETBio (left) and the corresponding model cone density (right) (compare to Fig 1B). At bottom left, a comparison of the median diameter of cones are compared between the model and the M3 data. At bottom middle and bottom right, the cone diameters of the model and measured data from M3 are

compared across the horizontal and vertical meridians respectively.
(PDF)

**S8 Fig. Model cross-validation.** RMS errors for the 4 modeling scenarios we considered are depicted for 12 cells. Yellow and blue bars indicate insample and out-of-sample performance, respectively. During in-sample performance assessment the model is trained and evaluated using data from the same recording session. During out-of-sample performance assessment, the model is trained in one session and evaluated using data from another session. The data do not have enough power to reveal a model with best generalizing (out-of-sample) performance. In a few cells, the 1-cone/0.00D residual defocus model can be ruled out as its performance is significantly worse than the remaining 3. A two sample t-test with unequal variance was used to test against the hypothesis that there is a significant difference in the mean RMS fit errors between 2 modeling scenarios.
(PDF)

**S9 Fig. Optimization of residual defocus for different RGCs.** For each cell the optimal residual defocus value for the single cone center and multi-cone center model scenarios was calculated to gauge variability from the chosen 0.067 D reported in the main text. For each cell, the RMSE is shown for various residual defocus values for both model scarios. Cells are labeled L1-11 or M1-4 according to whether we believed they were likely to contain an L cone or M cone at their center.
(PDF)

**S10 Fig. All four model scenario fits for the 15 L-M and M-L cone opponent RGCs from the center of the fovea of M3.** Four model scenarios were considered for all 15 cells: single cone centers with 0 D residual defocus, single cone centers with 0.067 D residual defocus, multi-cone centers with 0 D residual defocus, and multi-cone centers with 0.067 D residual defocus. As can be seen from the fits, the single cone center with 0.067 D defocus performs well for all 15 cells and produces center-surround structures that are consistent with what has been measured from physiology. Cells are labeled L1-11 or M1-4 according to whether we believed they were likely to contain an L cone or M cone at their center.
(PDF)

**S11 Fig. Simple center-surround model estimating percentage of achromatic midget RGCs.** A simple model cone mosaic using hexagonal packing was created in MATLAB for the centermost 2400 cones (radius of 25 cones from the foveal center). Each cone was randomly assigned to be an L cone (48%), M cone (48%), or S cone (4%). For this simple model, a midget RGC was assumed to connect to one cone at its receptive field center and 6 cones at its surround (immediately adjacent to the center cone). For each model midget RGC, the proportion of cones that were the same type as the center cone was calculated. RGCs with all 6 surround cones of the same type as the center were classified as 'true achromatic' (orange histogram), and RGCs with at least 5 surround cones of the same type as the center were classified as 'thresholded achromatic' (blue histogram). The entire model simulation was then repeated 10,000 times, with a different randomly assigned cone mosaic each time. The histograms above show the number of simulations for which each percentage of midget RGCs that were either true achromatic or thresholded achromatic occurred. Based on this simple model, truly achromatic midget RGCs might make up approximately 1.5% of total midget RGCs at the foveal center, while thresholded achromatic midget RGCs might make up approximately 9% of total midget RGCs at the foveal center.
(PDF)

## Acknowledgments

The authors would like to thank Amber Walker for performing animal anesthesia and monitoring during imaging, Deniz Dalkara for providing the viral vector used in M1, Max Snodderly for assistance with estimating the effect of macular pigment, Charles Granger for optical alignment assistance, Daniel Guarino and Martin Gira for maintaining the stereotaxic cart and other electronic components, as well as Aby Joseph, Jennifer Hunter, and Jesse Schallek for data analysis suggestions. We thank the vector core at the Perelman School of Medicine, University of Pennsylvania and the Genetically-Encoded Neuronal Indicator and Effector (GENIE) Project and the Janelia Research Campus of the Howard Hughes Medical Institute, specifically Vivek Jayaraman, Ph.D., Douglas S. Kim, Ph.D., Loren L. Looger, Ph.D., and Karel Svoboda, Ph.D.

## Author Contributions

**Conceptualization:** Tyler Godat, Sara Patterson, Juliette E. McGregor, David H. Brainard, William H. Merigan, David R. Williams.

**Data curation:** Tyler Godat, Nicolas P. Cottaris, Kendall Kohout, Keith Parkins.

**Formal analysis:** Tyler Godat, Nicolas P. Cottaris, Sara Patterson, Kendall Kohout.

**Funding acquisition:** Tyler Godat, David H. Brainard, William H. Merigan, David R. Williams.

**Investigation:** Tyler Godat, Juliette E. McGregor.

**Methodology:** Tyler Godat, Nicolas P. Cottaris.

**Project administration:** Tyler Godat, Juliette E. McGregor, William H. Merigan, David R. Williams.

**Resources:** Qiang Yang, Jennifer M. Strazzeri.

**Software:** Tyler Godat, Sara Patterson, Kendall Kohout, Keith Parkins, Qiang Yang.

**Supervision:** Tyler Godat, David H. Brainard, William H. Merigan, David R. Williams.

**Validation:** Tyler Godat, Nicolas P. Cottaris.

**Visualization:** Tyler Godat, Nicolas P. Cottaris, Kendall Kohout.

**Writing – original draft:** Tyler Godat.

**Writing – review & editing:** Tyler Godat, Nicolas P. Cottaris, Sara Patterson, Kendall Kohout, Juliette E. McGregor, David H. Brainard, William H. Merigan, David R. Williams.

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
