## [Decision Letter · Decision Letter 0]

11 Aug 2022

PONE-D-22-17429*In vivo* chromatic and spatial tuning of foveolar retinal ganglion cells in *Macaca fascicularis*PLOS ONE

Dear Dr. Godat,

Thank you for submitting your manuscript to PLOS ONE. After careful consideration, we feel that it has merit but does not fully meet PLOS ONE’s publication criteria as it currently stands. Therefore, we invite you to submit a revised version of the manuscript that addresses the points raised during the review process.

Please address all reviewers' comments in your review, in particular the requests for the raw data and visual aids explaining the methods.

We look forward to receiving your revised manuscript.

Kind regards,

Manuel Spitschan

Academic Editor

PLOS ONE

Journal Requirements:

I have read the journal’s policy and the authors of this manuscript have the following competing interests: D.R.W. and W.H.M. receive funding from the National Eye Institute. D.R.W. has received additional funding from The Arnold and Mabel Beckman Foundation, Alcon, and Warby Parker, and W.H.M. receives funding from Research to Prevent Blindness. D.H.B. receives funding from the National Institutes of Health, Johnson & Johnson, and Facebook Reality Labs. D.R.W. has patents with the University of Rochester for adaptive optics imaging of the retina: US patent #6,199,986 “Rapid, automatic measurement of the eye’s wave aberration”. US patent #6,264,328 “Wavefront sensor with off-axis illumination” and US patent 6,338,559 “Apparatus and method for improving vision and retinal imaging”. Q.Y. has patents with the University of Rochester, Canon Inc. and the University of Montana, for image stabilization algorithms: US patent #9,226,656: “Real-time optical and digital image stabilization for adaptive optics scanning ophthalmoscopy”, US patent # 9,406,133: “System and method for real-time image registration”, US patent #: 9,485,383, “Imaging based correction of distortion from a scanner” and US patent #:9,454,084, “Light source modulation for a scanning microscope”. D.H.B. is an inventor on US patent App. 16/389,942. Additionally. Q.Y. has undertaken consultancy work for Oculus VR and Boston Micromachine Corporation. These competing interests do not alter our adherence to PLOS ONE policies on sharing data and materials.  

We note that you received funding from commercial sources: National Eye Institute, Arnold and Mabel Beckman Foundation, Alcon, and Warby Parker, Johnson & Johnson, and Facebook Reality Lab.

Reviewers' comments:

Reviewer's Responses to Questions

**Comments to the Author**

1. Is the manuscript technically sound, and do the data support the conclusions?

Reviewer #1: Yes

Reviewer #2: Partly

2. Has the statistical analysis been performed appropriately and rigorously? 

Reviewer #1: N/A

Reviewer #2: I Don't Know

3. Have the authors made all data underlying the findings in their manuscript fully available?

Reviewer #1: Yes

Reviewer #2: Yes

4. Is the manuscript presented in an intelligible fashion and written in standard English?

Reviewer #1: Yes

Reviewer #2: Yes

5. Review Comments to the Author

Reviewer #1: In this study, Godat and colleagues investigate response properties from foveolar retinal ganglion cells (RGCs) in the non-human primate retina using in vivo functional imaging.

They combine adaptive optics ophthalmoscopy with calcium imaging to optically record the activity of RGCs virally transduced with calcium indicator. Indeed, the fovea structure makes it possible to stimulate foveolar cones while recording calcium responses evoked in the corresponding RGCs in two spatially distinct areas. Such an approach had recently been reported by the same group (McGregor et al, Nat com 2020) to evaluate a strategy of vision restoration via optogenetic therapy. In this manuscript, they demonstrate its interest in basic science.

First, they functionally sort the RGCs using chromatic stimuli. They separate putative midget RGCs, determine their spatial frequency function, and model their receptive field. Interestingly, they found higher spatial sensitivity compared to previous studies suggesting that measures collected through animal optics (cornea, aqueous, lens, and vitreous) may not faithfully represent the RGCs properties and/or that properties from more eccentric RGCs may differ.

Overall, this is an interesting manuscript. Importantly, the authors carefully report the limitations/uncertainties regarding the method and their modelization.

I am just providing here a few minor comments/questions that the authors may consider.

(1) The overall time course of the experiment should be clarified. What is the time window of the recordings after the AAV injection? Is there a potential long-term impact of the GCamp6 f/s on the RGCs? How stable is its expression?

(2) Did the author notice differences between retinas transduced with different calcium indicators (GCamp6f vs GCamp6s) or vectors?

(3) Time of the day, the overall length of the sessions as well as previous light history should be reported.

(4) Line 221, the FWHM of the Thorlabs LED could be mentioned.

(5) Which drug is used for pupil dilation?

(6) As mentioned, technical limitations of in vivo functional imaging are duly mentioned and discussed (along with the advantages of this method) in different sections of the manuscript (for example L170 in the methods or line 535 in the results). A suggestion would be to consolidate these remarks into a discussion paragraph comparing this relatively recent method to more classical electrophysiology techniques.

(7) Line 985, the abbreviation PSF does not seem to be defined.

(8) L1055 the conflicts of interest appear to be displaced in the references section.

(9) Figure 5, is “green points (squares), 640 blue points (triangles), and red points (circles)” the right temporal order? Did the authors notice any trend over the 3 weeks?

(10) Figure 7 e and f, there is no legend on the y-axis.

(11) Figure 9, the center/surround caption is very hard to read.

Reviewer #2: In this manuscript, Godat and colleagues recorded visual responses of retinal ganglion cells in the central fovea of living primates using calcium imaging. Recordings from these regions of highest acuity have been challenging in in-vitro studies due to the difficulties in preserving the fovea during retinal extraction and the displacement of retinal ganglion cells and their input photoreceptors. The authors confirm that previously described properties of more peripheral midget and parasol ganglion cells are conserved in the fovea. Furthermore, they provide a model to describe how retinal ganglion cells sample from the underlying cone photoreceptor mosaic. They use this model to demonstrate that their recordings are consistent with the hypothesis that midget ganglion cells in the central fovea sample from one or few cones to preserve the resolution of the underlying photoreceptor mosaic.

Our knowledge on foveal retinal processing is limited to anatomical analyses, and parafoveal and LGN measurements. This manuscript hence provides an interesting new data set and confirms hypotheses derived from previous studies. In particular, the model of cone mosaic and retinal ganglion cell responses allowed the authors to determine whether foveal retinal ganglion cells preserve the resolution of the underlying photoreceptor mosaic. For publication, three main points should be addressed: (1) The manuscript is difficult to understand for readers who are not experts in optics. It would profit from illustrations of the experimental setup. (2) The manuscript lacks figures showing raw data. (3) A concluding paragraph is missing.

Major comments:

(1) Provide visual help to understand methodology: Large parts of the manuscript are, understandably, very technical and only comprehensible to experts in optics. However, schematics illustrating the overall setup and the most important issues and corrections, together with clear conceptual explanations of the setup/measurements/analysis would make this manuscript much more accessible to a broader readership.

(2) Lack of raw data: The manuscript contains only one trace of raw data (Figure 2a). It is very difficult for the reader to judge the quality of the data, the repeatability of responses across trials and sessions, and to compare it to previously published in-vitro and LGN data. One of the apparent advantages of the here used technique is that the same cells can be imaged across multiple days. However, little data and no metrics are presented to illustrate how consistent these responses are. The authors should provide more raw traces that illustrate the different cell types, subjects, and repeated measurements, also for the color stimuli; and it would be interesting to see some metrics on the within- and between-session consistency of the responses (e.g. of the average curves shown in Fig. 5).

(3) Concluding paragraph: Currently, confirmation of previous findings and novel or unexpected findings are distributed across various paragraphs. A concluding paragraph highlighting which hypotheses/findings from LGN recordings and anatomical studies were confirmed, and which novel aspects were found, would make the manuscript easier to understand.

Minor comments:

Technical information for primate researchers: Given that only very few primates can be used for invasive studies, all information that allows optimizing future studies is highly valuable. Therefore, it would be of service to the community if the authors could add a paragraph on the different conditions used in M1, M2 and M3. For instance, only M2 and M3 received cyclosporin A treatment; was it not necessary in M1 due to the different vector used to express GCaMP? How did the efficiency of transmission compare between 7m8-SNCG-GCaMP6f and AAV2-CAG-GCaMP6s? Was there a difference between the signal of the fast and slow GCaMP in the context of the described image acquisition?

Line 160: Can the authors clarify what the Watt measurements refer to? Is this the amount of light across the whole pupil?

Line 163: Please spell out the referenced maximum permissible exposure value.

Line 358: What is the assumption of 199 um/deg based on?

Line 1055: correct reference

Figure 5: The plots appear to be cut. Some dots below 0 df/f are not visible (e.g. L1) and probably also some error bars are cut (e.g. M2).

Figure 7e: typo in “single”

Figure 9a: Cell M4 has very variable responses across repetitions (Figure 5) and the here used average tuning curve is hence not very informative. Another example cell would be more suited to demonstrate the effects of the optics.

6. PLOS authors have the option to publish the peer review history of their article (what does this mean?). If published, this will include your full peer review and any attached files.

Reviewer #1: No

Reviewer #2: No

---

## [Author Response · Author response to Decision Letter 0]

16 Sep 2022

Dear Reviewers,

 We thank you for your time and attention and for the many insightful comments. Where possible, we have attempted to fully answer each of your criticisms or questions in the main body of the text or in the Supplement. Below please find your original comments in black and our answers in red (in the uploaded document entitled "Response to Reviewers". In this text box, all the text is black). 

Reviewer #1:

(1) The overall time course of the experiment should be clarified. What is the time window of the recordings after the AAV injection? Is there a potential long-term impact of the GCamp6 f/s on the RGCs? How stable is its expression?

We have clarified the timing of recordings in Manuscript in the “AAV mediated gene delivery to retina” Section of the Methods. M1 was imaged 2 years post-injection, M2 was imaged 1-1.3 years post-injection, and M3 was imaged 1.25-1.5 years post-injection. There have been studies that have reported GCaMP toxicity (e.g. https://www.biorxiv.org/content/10.1101/2022.01.09.475579v1.full ), but we feel that these considerations fall outside the scope of our current study as we only include cells that are responsive to the stimuli used. We did notice that there were some cells observable in the en face fluorescence images that did not respond to any of the chromatic or spatial stimuli used, and these could be cells that have succumbed to toxicity, but they are not reported in this study as they could also be rarer cell classes with specialized tuning. All of the data sets averaged from this study were taken within the time course of a single month except for data in M2 with a maximum separation of 4 months (data sets from M3 at two time points 3 months apart were taken at slightly different locations with different imaging parameters so direct expression comparisons are not possible), so we also feel it is beyond the scope of this study to comment on the long term stability of GCaMP expression. However, in animals whose data is not presented in this study, we have observed stable expression for periods of over two years. 

(2) Did the author notice differences between retinas transduced with different calcium indicators (GCamp6f vs GCamp6s) or vectors?

There were certainly differences between M1 (GCaMP6f) and M2/M3 (GCaMP6s). In general, one would expect the GCaMP6f to be faster and have lower sensitivity. The speed differences were not a factor in our study, as the 0.15-0.2 Hz frequencies used for the chromatic stimuli were well below the time constants of both GCaMP6f and 6s. We did however observe lower sensitivity and overall lower signal in M1 (noted in the manuscript), but animal M1 did not receive immune suppression prior to AAV injection (also noted in the manuscript), so it is possible that the lower signal in that animal is a combination of lower sensitivity from the GCaMP6f and worse viral infection rates (e.g. fewer cells infected and less GCaMP present in cells) from the lack of immune suppression. In any case, these possibilities are speculative. An understanding would require a large scale comparison between many animals with each vector, and we believe this is outside the scope of the current study. 

(3) Time of the day, the overall length of the sessions as well as previous light history should be reported.

Added information to “Adaptive optics imaging.” in the Methods. Imaging sessions started at 9 am and lasted between two to four hours. Previous light imaging history was limited to clinical fundus and SLO imaging (also described in “AAV mediated gene delivery to retina” in Methods). 

(4) Line 221, the FWHM of the Thorlabs LED could be mentioned.

FWHM added for each LED: 420±7 nm, 530±17 nm, and 660±10 nm

(5) Which drug is used for pupil dilation?

Added the drug used for pupil dilation into the text describing anesthesia in Methods (Tropicamide 1% and Phenylephrine 2.5%). The full list of medications and anesthesia protocols is found in this reference as referenced in the manuscript. (https://www.nature.com/articles/s41467-020-15317-6) 

(6) As mentioned, technical limitations of in vivo functional imaging are duly mentioned and discussed (along with the advantages of this method) in different sections of the manuscript (for example L170 in the methods or line 535 in the results). A suggestion would be to consolidate these remarks into a discussion paragraph comparing this relatively recent method to more classical electrophysiology techniques.

Added a new section to the Discussion entitled “Comparison of methodology used in this study with traditional electrophysiology”. Moved all of the comments noted from L535 to this new section, and also reiterated and expounded upon comments such as those noted in L170.

(7) Line 985, the abbreviation PSF does not seem to be defined.

Changed to point spread function.

(8) L1055 the conflicts of interest appear to be displaced in the references section.

There was an issue with the EndNote reference file, which has been fixed now.

(9) Figure 5, is “green points (squares), 640 blue points (triangles), and red points (circles)” the right temporal order? Did the authors notice any trend over the 3 weeks?

Clarified in the text, red points are Week 1, blue points are Week 2, and green points are Week 3. The image quality of the eye was slightly better in Week 3, and you can see for some cells (but not all) that the green points had the highest values, but overall we didn’t notice any consistent trend for all cells across all 3 weeks. 

(10) Figure 7 e and f, there is no legend on the y-axis.

Fixed, the y-axis is now labeled as residual error to match the text of the caption.

(11) Figure 9, the center/surround caption is very hard to read.

Fixed, made the captions much larger and legible.

Reviewer #2: 

Major comments:

(1) Provide visual help to understand methodology: Large parts of the manuscript are, understandably, very technical and only comprehensible to experts in optics. However, schematics illustrating the overall setup and the most important issues and corrections, together with clear conceptual explanations of the setup/measurements/analysis would make this manuscript much more accessible to a broader readership.

To address this comment, we have added an entirely new Supplemental figure “S1 Fig” entitled ‘AOSLO System Diagram’. This Supplemental figure shows the entire system used for stimulation and recording and describes the basics of how the imaging is done. We feel that full details of the control electronics and optical design of the system should be left to the cited references, but hope that this new diagram and description provides enough context for a broader readership. 

(2) Lack of raw data: The manuscript contains only one trace of raw data (Figure 2a). It is very difficult for the reader to judge the quality of the data, the repeatability of responses across trials and sessions, and to compare it to previously published in-vitro and LGN data. One of the apparent advantages of the here used technique is that the same cells can be imaged across multiple days. However, little data and no metrics are presented to illustrate how consistent these responses are. The authors should provide more raw traces that illustrate the different cell types, subjects, and repeated measurements, also for the color stimuli; and it would be interesting to see some metrics on the within- and between-session consistency of the responses (e.g. of the average curves shown in Fig. 5).

To address this, we have added two entirely new Supplemental figures:

a) S2 Fig “Examples of raw fluorescence time courses of putative midget cells to various stimuli”. This supplemental figure includes raw time courses from six of the L vs M cone opponent putative midget cells (e.g. as shown in Fig 5). For each cell shown, responses are shown for the L isolating, M isolating, and S isolating chromatic stimuli, as well as for the 9.2 c/deg, 28.3 c/deg, and 49.1 c/deg drifting gratings (to showcase a ‘low’, ‘medium’, and ‘high’ spatial frequency response). For each individual stimulus for a given cell, there are three time courses (blue, orange, yellow) that show the response of that cell from each of the three experiments performed, and the average time course shown in black. These cells were chosen to showcase examples of cells with both high and low delta F over F responses to the drifting gratings, and do not represent the “best” cells. We hope that the reviewers can appreciate the challenge of presenting the raw responses of 300+ cells to almost 20 different stimuli across multiple experiments and animals and agree that this subset of the key putative midget cells from the manuscript is a satisfactory representation. Note also that we have uploaded the minimal data set (which includes all raw time courses for all cell responses to every stimulus) to https://osf.io/s9qw4/ which is freely available and will be noted in the Data Availability section of the Manuscript.

b) S3 Fig “Variability of responses to chromatic and achromatic flicker”. This supplemental figure includes the variability in responses to the L isolating, M isolating, S isolating, and Luminance stimuli for all fifteen putative midget cells from Fig 5. Additionally, the variability of the luminance only cells, L cone only cells, and M cone only cells is plotted for the small field of view condition in animal M3. For the large field of view conditions in M3 and M2, there were too many cells to plot individually, so histograms are shown for each stimulus that plot the number of cells with a given standard error (divided by SNR) between repeated measurements. Note that for the box plots from the small field of view condition in M3, there were only 3 measurements, so the median point and both edges of the whiskers for each box represent the three actual measurements for that cell. No variability data is presented for animal M1 as there was only one experiment in that animal. No data for variability within a single experiment is presented because the noise floor is only overcome for the majority of cells when trials are averaged—this is a combination of sparse fluorescence signal from the GCaMP6 and the fact that the cone modulations are low for stimuli that affect the most cells (less than 20% for L and M isolating stimuli as noted in the manuscript).

(3) Concluding paragraph: Currently, confirmation of previous findings and novel or unexpected findings are distributed across various paragraphs. A concluding paragraph highlighting which hypotheses/findings from LGN recordings and anatomical studies were confirmed, and which novel aspects were found, would make the manuscript easier to understand.

As addressed in a comment from the other reviewer, a new Discussion paragraph entitled “Comparison of methodology used in this study with traditional electrophysiology” has been added to address part of this criticism. We have taken comments that were spread out across various paragraphs, consolidated them here, and attempted to enhance understanding of how this manuscript relates to the previous recordings. We have also added a concise Conclusions paragraph to address which findings were confirmed and which were novel. 

Minor comments:

Technical information for primate researchers: Given that only very few primates can be used for invasive studies, all information that allows optimizing future studies is highly valuable. Therefore, it would be of service to the community if the authors could add a paragraph on the different conditions used in M1, M2 and M3. For instance, only M2 and M3 received cyclosporin A treatment; was it not necessary in M1 due to the different vector used to express GCaMP? How did the efficiency of transmission compare between 7m8-SNCG-GCaMP6f and AAV2-CAG-GCaMP6s? Was there a difference between the signal of the fast and slow GCaMP in the context of the described image acquisition?

Added additional information to “Immune Suppression” and “AAV mediated gene delivery to retina” sections in the Methods. In general, it is difficult to draw conclusions about what the best practices are since we didn’t explicitly test for that experimentally, and PLOS ONE has strict policies about referring to data that is not shown in the manuscript, but we have tried to make more clear the reasoning behind the different choices made for the animals. M1 did not receive immune suppression because the group wasn’t sure if it was necessary at the time (and the immune suppression is invasive). Consequently, M1 did have the worst expression and signal of the three animals, but it is impossible to separate out the effect of immune suppression from any other reasons for differences in the vector transmission efficiency in this case since the differences occurred only in one animal. Our current practice is to immune suppress the animals for several weeks prior to injection as we believe that produces the best expression of the vectors in the cells. As noted in our response to a similar comment from Reviewer 1 above, the speed of the chromatic stimuli (0.15 – 0.2 Hz) was chosen to be within the time constants of both GCaMP 6s and 6f, so there was no temporal difference in the acquisition of the signals from the different animals. 

Line 160: Can the authors clarify what the Watt measurements refer to? Is this the amount of light across the whole pupil?

Clarified in the text by explicitly stating that it is optical power measured across the entire pupil—the amount of optical power that enters the animal’s eye. 

Line 163: Please spell out the referenced maximum permissible exposure value.

Maximum permissible exposure values are calculated according to the ANSI standard as summed ratios that end up needing to be less than unity. This has been clarified in the text and the range of exposure values used has also been listed. The actual maximum exposure value in terms of light power is dependent on the length of time the source was presented, and so is different for every different experiment (hence the normalization to unity limit which allows comparison across experiments). 

Line 358: What is the assumption of 199 um/deg based on?

Fixed to be explicitly stated in the text. The axial length of M3 is 16.56 mm, and the model human eye has an axial length of 24.2 mm and conversion of 291.2 μm/deg. Thus, you get 16.56/24.2 * 291.2 ≈ 199 μm/deg

Line 1055: correct reference

Fixed reference. 

Figure 5: The plots appear to be cut. Some dots below 0 df/f are not visible (e.g. L1) and probably also some error bars are cut (e.g. M2).

Fixed plot limits so all dots and error bars are visible.

Figure 7e: typo in “single”

Fixed typo. 

Figure 9a: Cell M4 has very variable responses across repetitions (Figure 5) and the here used average tuning curve is hence not very informative. Another example cell would be more suited to demonstrate the effects of the optics.

Cell M4 replaced with cell L11 which has much more consistent responses across repetitions.

---

## [Decision Letter · Decision Letter 1]

14 Nov 2022

*In vivo* chromatic and spatial tuning of foveolar retinal ganglion cells in *Macaca fascicularis*

PONE-D-22-17429R1

Dear Dr. Godat,

We’re pleased to inform you that your manuscript has been judged scientifically suitable for publication and will be formally accepted for publication once it meets all outstanding technical requirements.

Kind regards,

Manuel Spitschan

Academic Editor

PLOS ONE

Additional Editor Comments (optional):

Reviewers' comments:

Reviewer's Responses to Questions

**Comments to the Author**

1. If the authors have adequately addressed your comments raised in a previous round of review and you feel that this manuscript is now acceptable for publication, you may indicate that here to bypass the “Comments to the Author” section, enter your conflict of interest statement in the “Confidential to Editor” section, and submit your "Accept" recommendation.

Reviewer #1: All comments have been addressed

Reviewer #2: All comments have been addressed

2. Is the manuscript technically sound, and do the data support the conclusions?

Reviewer #1: Yes

Reviewer #2: Yes

3. Has the statistical analysis been performed appropriately and rigorously? 

Reviewer #1: Yes

Reviewer #2: I Don't Know

4. Have the authors made all data underlying the findings in their manuscript fully available?

Reviewer #1: Yes

Reviewer #2: Yes

5. Is the manuscript presented in an intelligible fashion and written in standard English?

Reviewer #1: Yes

Reviewer #2: Yes

6. Review Comments to the Author

Reviewer #1: The authors have answered all my questions and requests for clarification. The manuscript is now ready for publication.

Reviewer #2: I would like to thank the authors for providing the additional raw data representations, analysis, and technical details. All my comments have been addressed and the manuscript has gained in clarity and readability for a broader readership. I fully support publication of the revised manuscript and am sure the retina field will appreciate this very difficult to obtain set of data and analysis.

7. PLOS authors have the option to publish the peer review history of their article (what does this mean?). If published, this will include your full peer review and any attached files.

Reviewer #1: No

Reviewer #2: No

---

## [Editor Report · Acceptance letter]

17 Nov 2022

PONE-D-22-17429R1 

*In vivo* chromatic and spatial tuning of foveolar retinal ganglion cells in *Macaca fascicularis*

Dear Dr. Godat:

I'm pleased to inform you that your manuscript has been deemed suitable for publication in PLOS ONE. Congratulations! Your manuscript is now with our production department. 

Kind regards, 

on behalf of

Dr. Manuel Spitschan 

Academic Editor

PLOS ONE